# Past and Future Responses of Soil Water to Climate Change in Tropical and Subtropical Rainforest Systems in South America

Santiago M. Márquez Arévalo [1], Rafael Coll Delgado [2,*], Douglas da Silva Lindemann [3], Yuri A. Gelsleichter [4], Marcos Gervasio Pereira [5], Rafael de Ávila Rodrigues [6], Flávio Barbosa Justino [7], Henderson Silva Wanderley [2], Everaldo Zonta [5], Romário Oliveira de Santana [8] and Renato Sinquini de Souza [5]

[1] Graduate Program in Applied Meteorology, Federal University of Viçosa (UFV), Viçosa 36570-900, Brazil; santiago.arevalo@ufv.br

[2] Department of Environmental Sciences, Forest Institute, Federal Rural University of Rio de Janeiro (UFRRJ), Seropédica 23897-000, Brazil; henderson@ufrrj.br

[3] Faculty of Meteorology, Federal University of Pelotas, University Campus, S/N, Building 110, Capão do Leão 96160-000, Brazil; douglas.lindemann@ufpel.edu.br

[4] Department of Soil Science, Institute of Environmental Sciences, Hungarian University of Agriculture and Life Sciences, Páter Károly u. 1, H-2100 Gödöllő, Hungary; gelsleichter.yuri.andrei@uni-mate.hu

[5] Department of Soils, Federal Rural University of Rio de Janeiro (UFRRJ), Seropédica 23897-000, Brazil; gervasio@ufrrj.br (M.G.P.); ezonta@ufrrj.br (E.Z.)

[6] Special Academic Unit Institute of Geography, Federal University Catalão, Goiás 75704-020, Brazil; rafael.rodrigues@ufcat.edu.br

[7] Department of Agricultural Engineering—Room 121A, Federal University of Viçosa (UFV), Viçosa 36570-900, Brazil; fjustino@ufv.br

[8] Department of Economics, State University of Santa Cruz (UESC), Ilhéus 45662-000, Brazil; rosantana@uesc.br

* Correspondence: rafaelcoll@ufrrj.br

**Abstract:** The present study aimed to contribute to the diagnosis and advance the knowledge of the impacts of land use change and climate change on the tropical longleaf forest biome at the continental scale in South America (Biome 1 according to the WWF classification) for realizing scientific progress in the search for convincing strategies and actions by different actors for the preservation of forests in the continent. The status and climate of the area, which harbors the tropical longleaf forests of South America, were assessed. Moreover, volumetric soil moisture (VSM) was evaluated through maps and simulation using the autoregressive integrated moving average model (ARIMA). Furthermore, future climate scenarios were predicted based on El Niño–Southern Oscillation phenomena, meteorological systems, and scientific evidence, such as the shared socioeconomic pathways (SSPs) and sociopolitical dynamics evident in the region from the case analysis of the Brazilian states of Acre and Rio de Janeiro. An increase was noted in the temperature and range of precipitation variation in the biome. ARIMA analysis indicated changes of up to 0.24 m$^3$ m$^{-3}$ and an increased range of future VSM values. The December–January–February (DJF) quarter recorded the highest VSM median with the measurement scale of 0.05 to 0.44 m$^3$ m$^{-3}$, while the June–July–August (JJA) quarter recorded the lowest value. The regions of the biome with the lowest VSM values included southern Amazon (Ecuador, Peru, and the Brazilian states of Acre, Mato Grosso, Pará, and Maranhão), Brazilian Atlantic Forest, Southeast Region, and the Brazilian state of Bahia.

**Keywords:** biomes; drought vulnerability; future simulation; ARIMA analysis

## 1. Introduction

The Amazon and Atlantic Forest, referred to as Biome 1, "Broad Leaf Tropical and Subtropical Humid Forests", by the World Wide Fund for Nature (WWF) classification, is considered the richest biome in the world, covering over half of the territory of South America (AS) and occupying over 60% of the Brazilian territory [1]. These biomes are dynamically influenced by soil water, which controls vegetation growth and death. The

native vegetation cover in these ecosystems maintains the water balance through evapotranspiration, infiltration, and soil runoff [2,3].

The Amazon is home to one-third of all species in the world. However, in recent decades, deforestation has increased considerably. Specifically, the Atlantic Forest has witnessed a greater degree of degradation, which started in colonial times, linked to the development of what Brazil is today [4]. Solid scientific evidence supports that a representative part of these ecosystems is being affected by global warming, driven primarily by anthropogenic action, particularly the expansion of extractive activities that damage the environment, the increasing frequency and severity of forest fires and a loss of vegetation in these regions [5,6].

Regarding climate change, soil water has been cataloged as one of the 50 most essential variables for the study of climate and climate change [7]. Therefore, understanding the effects of climate and land use on soil water balance is essential for the development of water resource management strategies in agricultural and environmental sciences [8]. For instance, the conversion of forests into agricultural land has considerably increased the frequency of forest fires in these regions, which has altered the properties of soil and possibly its chemical composition, leading to the development of hydrophobic soils [9,10].

Remote sensing has been used in these biomes as a viable tool for environmental, climate, and vegetation studies. The knowledge of meteorological systems operating in these biomes and their correlation with soil water is of paramount importance, as in the case of the El Niño–Southern Oscillation (ENSO) phenomenon, which alters the rainfall and water storage regimes in these biomes [11].

The effects of ENSO phenomena vary according to the region of the continent, depending on the interaction between the Andes Mountains located on the coast of the Pacific Ocean [12,13] and climate systems acting on the continent, such as the Bolivian Alta (AB), which has a strong ability to make air rise that is directly related to the rainfall regime of the South American continent [14,15]; the Intertropical Convergence Zone (ITCZ), which is the area where the northern hemisphere trade winds converge with the southern hemisphere trade winds; and the Humidity Convergence Zone (ZCOU), where the humidity and upward movements of air converge, forming clouds, rains, storms, and hurricanes [16].

In these regions, studies of the various factors affecting the environment depend on the construction of future scenarios [6,17]. Such information can be gathered for regions with large territorial extent, such as the Amazon and Atlantic Forest, thanks to the use of remote sensing, statistical models, and time series analyses.

To this end, the present study was aimed at understanding the association of geoclimatic and land use factors with soil water distribution patterns to fill the knowledge gap on the proposed subject and to unveil the effects of climate change on these biomes in South America. Our findings will aid the formulation of strategic growth plans and the rational use of natural resources in the Amazon and Atlantic Forest, achieving economic and social development of these regions while simultaneously preserving their environmental heritage, which in turn is associated with scientific and technological progress integrating economic advances with environmental protection.

## 2. Materials and Methods

### 2.1. Study Area

The broadleaf tropical and subtropical moist forest biome classified as Biome 1 by WWF in South America covers approximately half the area of the continent and is present in all its countries, with the exception of Chile and Uruguay, spanning from $10°42'7''$ N and $33°43'1''$ S to $34°43'37''$ W and $80°28'55''$ W.

This biome is composed of two areas separated by Cerrado and Caatinga, between which humidity and biotic exchange occur. To use practical terminology covering the two large areas of forest that predominate in the continent, in the present study, the denomination of predominantly equatorial regime forests (FRPEs) was selected to refer to the forests of the Amazon and the north of the continent, and predominantly tropical

forests (FRPTs) were to refer to the group of forests composed mainly of the Atlantic Forest near the Tropic of Capricorn (Figure 1).

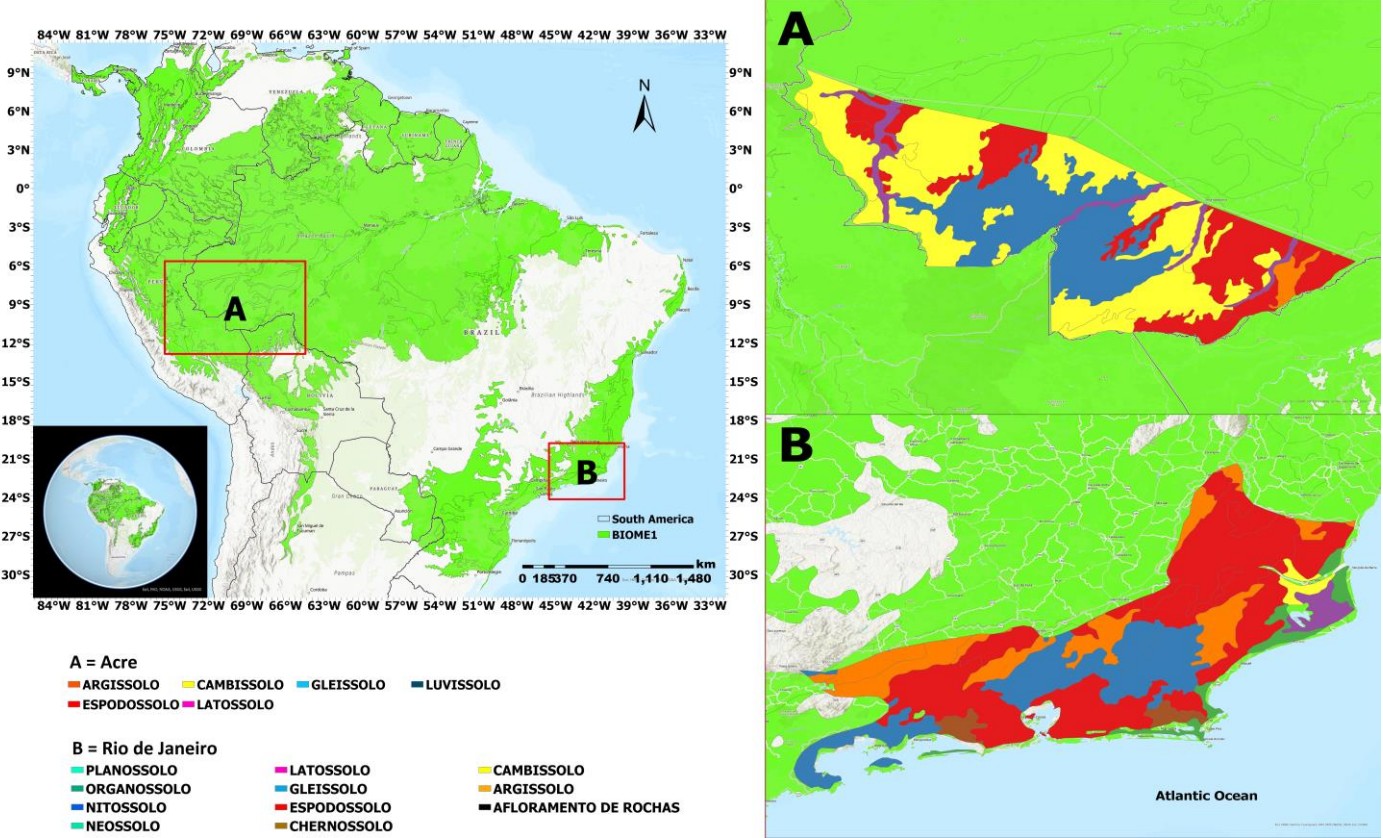

**Figure 1.** Coverage area and geographic location of Biome 1 in South America. (**A**) EAC and (**B**) ERJ with the predominant soil classification in these regions.

To offer a perspective on the local scale, the territorial units of the Brazilian states of Acre (A), with an area of 164,123,964 km$^2$, and Rio de Janeiro (B), with an area of 43,750,427 km$^2$, according to data from the Brazilian Institute of Geography and Statistics, were selected to analyze the number of fire foci and the volumetric soil moisture (VSM) (m$^3$ m$^{-3}$) relative to the existing soil units in each state (Figure 1), according to the classification of the Brazilian System of Classification of Soil (SiBCS) [18].

The State of Acre (EAC) is located in the Brazilian Amazon forest, bordering the Brazilian state of Amazonas in the north and east and bordering Bolivia and Peru in the south and west. The altitude varies between 200 and 300 m, with the exception being Serra de Contamara on the west, where the altitude is as high as 609 m. Owing to its location, this territory is influenced by its proximity to the Andes Mountains and the continental nature of its climate regime.

The State of Rio de Janeiro (ERJ) borders the Atlantic Ocean in the south and southeast, the state of Espírito Santo in the northeast, the state of Minas Gerais in the northwest, and the state of São Paulo in the southwest. The altitude in this state varies from sea level to the maximum altitude of 2791 m above sea level at Pico das Agulhas Negras.

*2.2. Past Meteorological Information—Climatic Research Unit Dataset (CRU)*

In order to have a general estimate of the climate of the entire biome studied, the variables air temperature 2 m above the soil surface (°C) and precipitation (mm) were considered on the monthly scale for the time series 2001–2019; the series of data was obtained from the Copernicus Data Store platform in this temporal range [19] in the NetCDF-4 format, with a spatial resolution of 0.5° × 0.5° [20].

From NetCDF-4, a grid cell was created to extract air temperature and precipitation data for the area of Biome 1 using the "extact by mask" tool. A rain and temperature boxplot analysis was performed, generating the boxplot graph in the R 4.1.1 software [21], using the "ggplot2" [22] and "tidyverse" [23] libraries.

### 2.3. Weather Database of the Future—CMIP6

Future meteorological data on precipitation (Kg m$^2$ s$^{-1}$) and air temperature at 2 m above ground surface (K) were extracted from the CMIP6 Climate Projections platform [24] for the available monthly time series from January 2015 to December 2050.

Two Shared Socioeconomic Pathways (SSPs) scenarios were used, SSP2-4.5, "SSP2: Halfway", and SSP5-8.5, "SSP5: Development driven by fossil fuels", from the Japanese model MIROC6 [25], with a spatial resolution of ~2.25° × ~2.25° in the NetCDF-4 format.

Precipitation data were obtained from the Copernicus Climate Data Store [26] and transformed to a monthly scale in mm units, multiplying each value by 86,400 to transform a day into seconds and multiplying that by the number of days in that month, and the values of temperature were converted to degrees Celsius (°C). In the R 4.1.1 software [21], a boxplot graph was generated on a monthly scale (*x* axis) for all years of the data series for the analysis.

### 2.4. Remote Sensing Data (Fire Foci)

In order to analyze the state of the biome, data from fire foci downloaded on a monthly scale for the years 2001–2021 were also obtained in a vector shapefile format (.SHP), using the reference sensor Moderate Resolution Imaging Spectroradiometer (MODIS) from the Fire Information for Resource Management System (NASA-FIRMS) [27] for the entire area of Biome 1 and for local analysis in the states of Acre and Rio de Janeiro.

Using the MCD14ML product, the MODIS sensor was operated with the AQUA and TERRA satellites at a spatial resolution of 1 × 1 km. Using the ArcGIS and spreadsheet software, fire foci were counted for the entire biome on a monthly scale, and with these data a stacked bar chart was constructed for each year of the study period.

### 2.5. Volumetric Soil Moisture (VSM) and Autoregressive Integrated Moving Average Model—ARIMA

The focus of this study was to carry out VSM modeling for the entirety of Biome 1 through the autoregressive integrated moving average (ARIMA) statistical model, defined by Equations (1) and (2).

$$X_t - \alpha_1 X_{t-1} - \cdots \alpha_{p'} X_{t-p'} = \varepsilon_{t-1} + \theta_{1\varepsilon_{t-1}} + \cdots + \theta_{q\varepsilon_{t-q}}, \tag{1}$$

or the equivalent:

$$(1 - \sum_{i=1}^{p'} \alpha_i L^i) X_t = (1 + \sum_{i=1}^{q} \theta_i L^i) \varepsilon_t \tag{2}$$

where $L$ is the lag operator, the $\alpha_i$ values are the parameters of the autoregressive part of the model, the $\theta_i$ values are the parameters of the moving average part, and the $\varepsilon_t$ values are error terms. The error terms ($\varepsilon_t$) are generally assumed to be independent, identically distributed variables sampled from a normal distribution with a mean of zero.

With VSM data (m$^3$ m$^{-3}$) taken by remote and passive sensors at a spatial resolution of 0.25° × 0.25° obtained from the Copernicus Data Store website [28] for the past period from 01/2001 to 12/2021 in a monthly format, after the delimitation of the Biome 1 area for AS, the average of the values per climatic quarter was calculated (DJF = December–January–February, MAM = March–April–May, JJA = June–July–August, and SON = September–October–November) of the entire time series, and with the results a single map was generated for each quarter.

The future VSM results were obtained from the ARIMA modeling carried out in the R software for the future period 2022–2050 from the VSM data from the period 2001–2021 using the "netcdf4" library. With the results of the future modeling, maps were generated

following the format of the past period, and a map was generated with the subtraction of the VSM values of the past series and the results of the future ARIMA modeling.

For the global and local analysis of the States of Acre and Rio de Janeiro, a grid cell was created with the same spatial resolution of the VSM data in a raster format, with which the VSM values were obtained for each order of soil in the two states analyzed for the past and future period, and the results were arranged in the form of boxplot graphs.

### 2.6. Statistical Analysis and Correlation Analysis between Variables

Using the R software, the VSM data were verified using the Mann–Kendall and Pettit tests. The Mann–Kendall test, described as a succession of $x_i$ of $n$ terms ($1 \leq i \leq n$) for a time series, $x = x_1, x_i, x_j, \dots, x_n$ [29,30], was applied to the time series of VSM in the period 2001–2021 on a monthly scale to identify data behavior, considering a value of $\alpha = 5\%$. Pettit's non-parametric test detects the breaking moment of a series; when performing two divisions of original non-stationary series, obtaining different distributions and means [31,32], the null hypothesis ($H_0$) assumes that there is no break in the series, while the alternative hypothesis ($H_1$) states that there is rupture [33]. Through the application of this test, a possible sign of rupture was found in the series of past VSM data (2001–2021) for the year 2007; for this year, VSM maps were generated on a quarterly scale from the remote sensing images taken by satellite.

Validation of future VSM data obtained through ARIMA modeling was carried out by calculating the statistical linear correlation coefficient (R), coefficient of determination ($R^2$), standard error of the estimate (EPE) ($m^3 \, m^{-3}$) and Willmott coefficient (D) [34] between this data series and the remote sensing VSM data.

To find a relationship between the variables in this study, a correlation matrix was performed in the R 4.1.1 software [21] using Pearson's correlation coefficients, denominated "r" between the variables evaluated in this work (fire foci, VSM, precipitation and air temperature), and the classifications of intensity with which the ENSO phenomenon occurred during the study period 2001–2021 were based on the Oceanic Niño Index (ONI) [35]; three classifications were for La Niña ("Weak" La Niña (WL), "Moderate" La Niña (ML), and "Strong" La Niña (SL)), and four classifications were for El Niño ("Weak" El Niño (WE), "Moderate" El Niño (ME), "Strong" El Niño (SE), and "Very Strong" El Niño (VSE)) (Table 1), the results of this analysis are presented in a figure format.

**Table 1.** El Niño and La Niña years and intensities.

| El Niño | | | La Niña | | |
|---|---|---|---|---|---|
| **Weak—11** | **Moderate—2** | **Very Strong—1** | **Weak—6** | **Moderate—3** | **Strong—2** |
| 2004–2005 | 2002–2003 | 2015–2016 | 2000–2001 | 2011–2012 | 2007–2008 |
| 2006–2007 | 2009–2010 | | 2005–2006 | 2020–2021 | 2010–2011 |
| 2014–2015 | | | 2008–2009 | 2021–2022 | |
| 2018–2019 | | | 2016–2017 | | |
| | | | 2017–2018 | | |

The number to the right of each intensity rating represents the number of periods the event has been observed in during the 2001–2021 series.

## 3. Results

### 3.1. Climate of the "Tropical Large Leaf Forest" Biome (Biome 1)

Almost all years in the data series recorded the highest monthly values during August and September. In 2004 (September, 147,664), 2005 (August, 148,834), 2007 (September, 173,500), and 2010 (September, 121,958), the highest values of fire foci per month were observed in the analyzed data series (Figure 2). The years 2009 (September, 43,447), 2013 (September, 46,106), and 2014 (September, 47,119) showed the lowest values in the data series for these months (Figure 2). The last three years closest to the current period, which showed the highest values for this variable, were as follows: August 2010 (114,950), 2019

(76,911), and 2021 (74,076) and September 2010 (121,958), 2017 (88,845), and 2020 (81,688) (Figure 2).

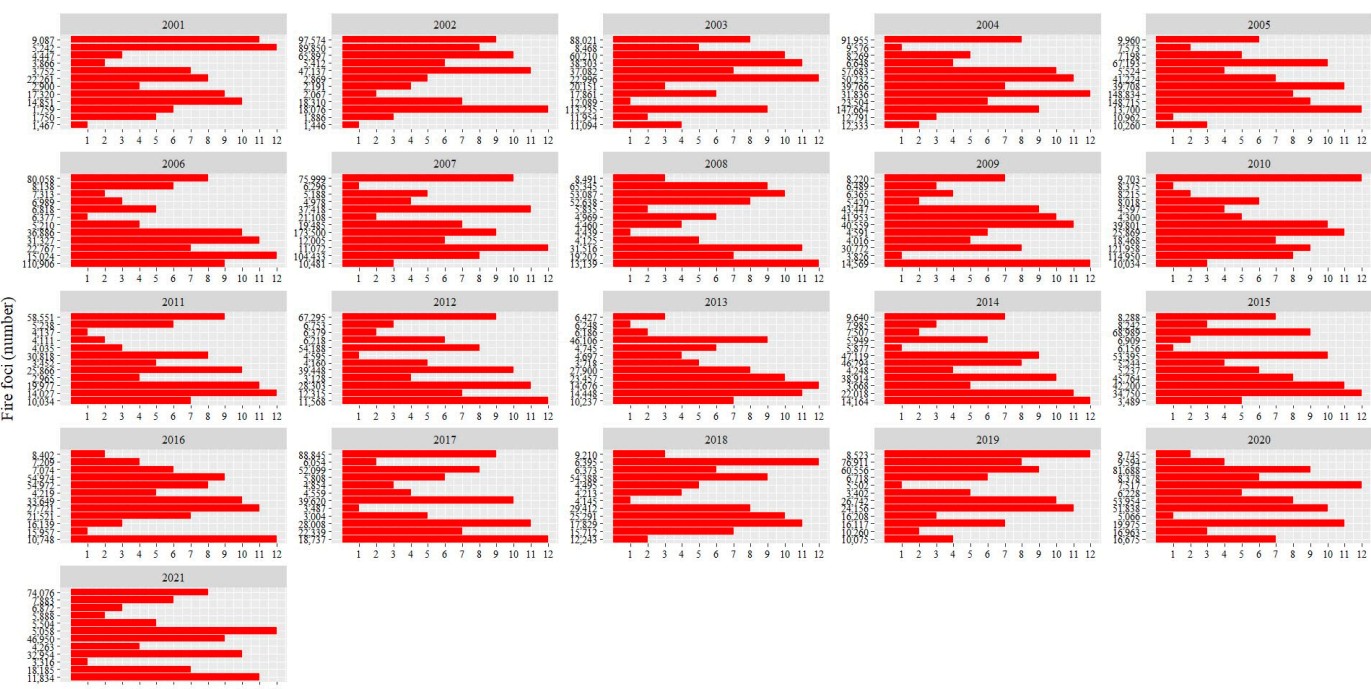

**Figure 2.** Temporal distribution of fire foci in the tropical and subtropical longleaf forest biome of South America in the 2001–2021 series.

The years 2013 and 2014 were predominantly in the neutral phase. Moreover, 2009 was a notable year, as it started above the "weak" La Niña threshold and later shifted into the "strong" El Niño phase. Even during this phase, fewer fire foci were recorded. The year 2008 showed the smallest distribution range of fire foci; this year started in the "strong" La Niña phase, shifted to the neutral phase over time, and ended up above the "weak" La Niña phase.

The maximum and minimum precipitation values were 266.2 mm (March 2009) and 91.44 mm (August 2001), respectively (Figure 3). The years 2004, 2009, and 2015 showed a positive distribution, indicating that most of the values were concentrated in the smallest fraction of quartile 2 in the boxplot; meanwhile, 2015 presented the smallest Q2 interval of precipitation in the time series.

For the years 2009, 2010, and 2011, high variance was noted in the precipitation values; as such, 2009 started in the "weak" La Niña phase and passed into the El Niño phase, reaching the "strong" El Niño phase. Year 2010 started in the "strong" El Niño phase and concluded in the "strong" La Niña phase, which marked the beginning of 2011, a year that started in the "intermediate" La Niña phase and concluded in the "moderate" La Niña phase.

Regarding ambient temperature, varied dispersion bands were noted in the time series, highlighting peaks in the mean trend line (in red) for the years 2005, 2010, and 2015, during which an ENSO event occurred (Figure 3). In 2015, the highest mean ambient temperature was recorded during September (26.4 °C). Of note is that the El Niño episode in 2015 was "very strong". Moreover, this was also the only year in the time series that showed positive asymmetry in the boxplot graph. The lowest recorded temperature was 23.5 °C during July 2004 (Figure 3).

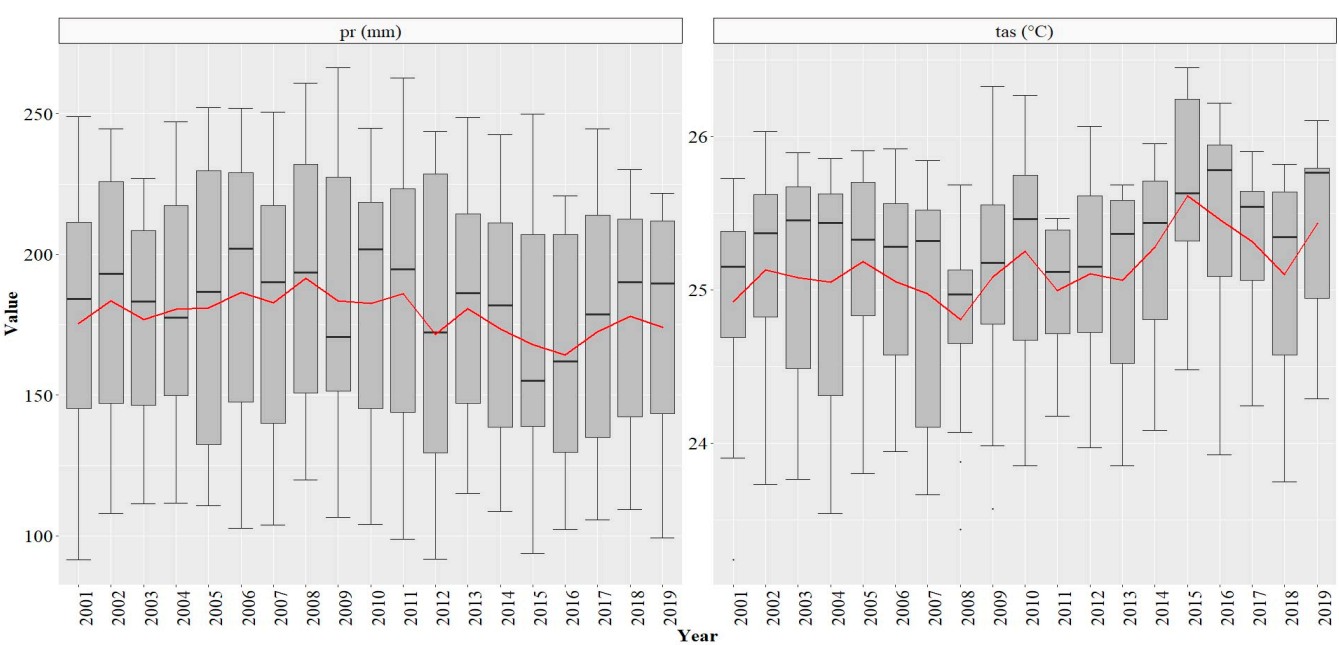

**Figure 3.** Boxplot statistical analysis: precipitation, pr (mm), in Biome 1 in the AS period 2001–2019, and air temperature, tas (°C), in Biome 1 in the AS period 2001–2019. The red line represents the mean of the time series.

*3.2. VSM Simulation*

In Figure 4, the VSM variation map generated through an autoregressive integrated moving average model (ARIMA) analysis of the observed data represents the highest values of soil water in dark green (0.44 m³ m⁻³) and the lowest values of soil water in orange (0.05 m³ m⁻³).

In the December–January–February (DJF) quarter, the lowest values of VSM were noted in the coastal regions of northeastern Brazil, corresponding to the states of Rio Grande do Norte, Paraíba, Alagoas, and Sergipe; the region comprising the border between Mato Grosso do Sul and São Paulo; the area around the mountainous zone on the triple border of Brazil, Guyana, and Venezuela, to the south of the latter country; and the forest of Paraguay on the border of Brazil.

Meanwhile, the highest values of VSM, represented by the green color on the map, were recorded in southern Brazil; part of the Southeast Region, with the exception of the interior of the state of São Paulo and the north of Espírito Santo, highlighting a patch between Mato Grosso and Pará in the east of the Brazilian Amazon (i.e., a major part of the Guianês massif, excluding the areas mentioned earlier with low VSM values); the Pacific coast of Colombia and northern Ecuador; and the forest of Bolivia.

During the March–April–May (MAM) quarter, the critical values of VSM were noted in the south of Venezuela. In Brazil, areas close to the border between the Brazilian states of Mato Grosso do Sul and São Paulo, and the regions north of Espírito Santo, northeast of Minas Gerais, and south of Bahia continued to exhibit similar trends to those in the previous quarter. A small increase in VSM values was observed on the northeastern coast of Brazil in the states of Rio Grande do Norte, Alagoas, and Sergipe. Higher values of soil water were recorded in areas close to the mouth of the Amazon River in the states of Pará and Guianas.

During the June–July–August (JJA) quarter, the Peruvian Andes and southeastern region of the Brazilian legal Amazon, encompassing the states of Mato Grosso, Maranhão, Tocantins, and Pará, showed low VSM values. The highest VSM values during this period were recorded at the northeastern coast because of rains occurring as a product of trade winds that reach the coast at this time of the year. During this quarter, the forest part located between Bolivia and northern Argentina showed the lowest VSM values.

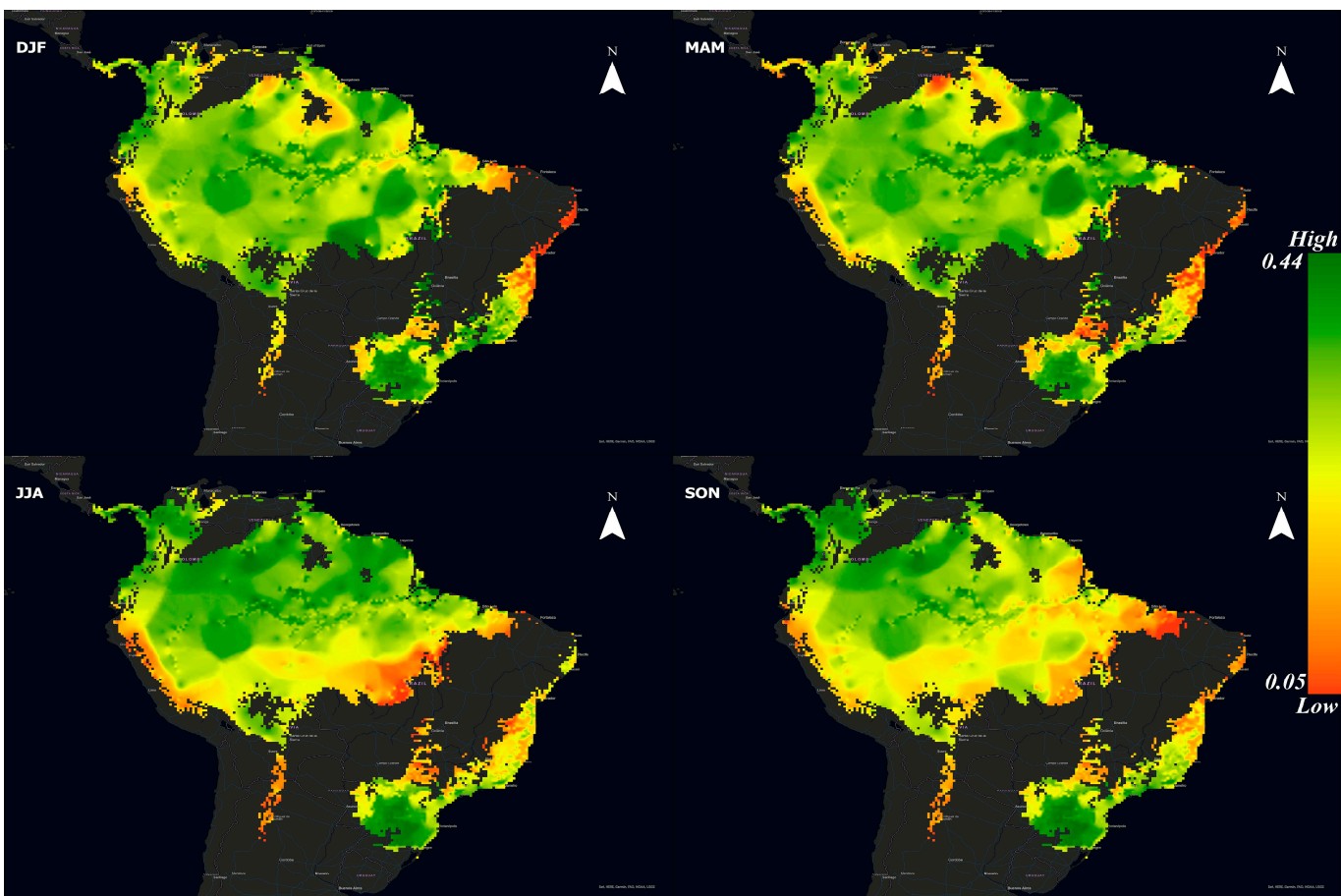

**Figure 4.** VSM spatial climate change (m$^3$ m$^{-3}$) for the 2001–2021 climate quarters.

In the September–October–November (SON) quarter, critical values were recorded in the Brazilian state of Maranhão, reaching the minimum on the measurement scale; important parts of the states of Pará and Amapá registered low VSM values. The Peruvian Andes region continued to show low VSM values.

Throughout the year, the forests located in southern Brazil and on the Pacific coast northwest of the continent showed high VSM values.

### 3.3. Analysis of Observed VSM and Climate Data for 2007

In the quarterly maps for 2007 generated using pixels extracted from netcdf4 images, the identification of water courses of the Amazon River Basin in the FRPE is highlighted, in which the area coverage of the biome reached a greater scope (Figure 5). The DJF quarter marked a period of extremely low values in the northeastern coastal regions of Brazil, with the exception of southern Bahia. In the MAM quarter, critical values were observed in the Brazilian states of São Paulo, an area with intense agricultural activities; the northeastern coast; and central Venezuela. Certain regions presented low VSM values during the JJA quarter, particularly the region between the Brazilian states of Mato Grosso, Pará, Tocantins, and Piauí as well as the region between Paraná and southeast Brazil. In addition, the Peruvian Andes recorded low values during this period. In the SON quarter, the Brazilian state of Piauí recorded the minimum value of the measurement scale (0.05 to 0.44 m$^3$ m$^{-3}$); likewise, the Southeast Region of Brazil showed low values of VSM, with red pixels in the states of São Paulo, Minas Gerais, and Espírito Santo, and the south of Bahia (a state in the Northeast Region) (Figure 5).

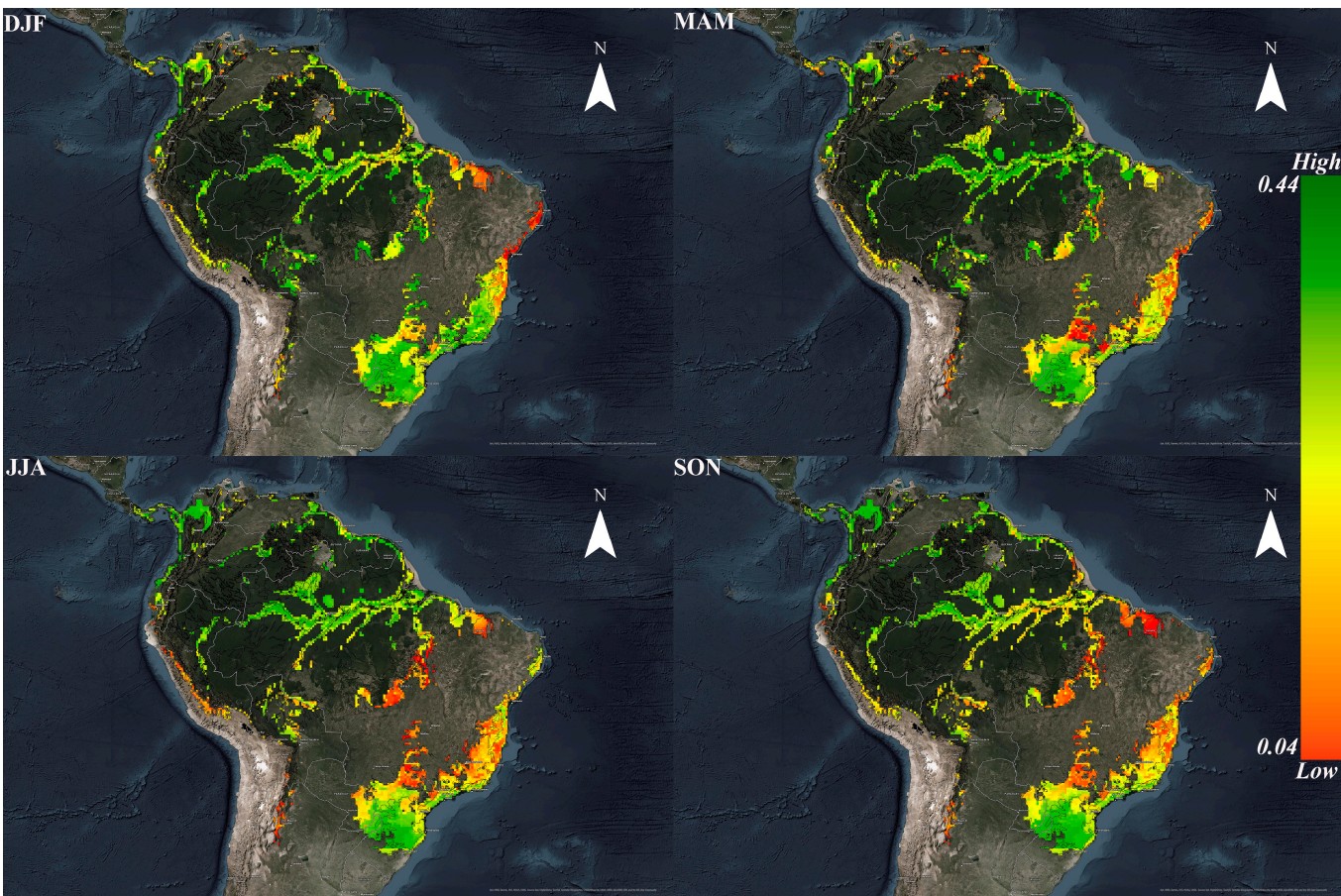

**Figure 5.** VSM spatial climate variation (m$^3$ m$^{-3}$) for 2007.

The meteorological systems that shape the regime of and variation in precipitation in the study region include the South Atlantic convergence zone (SACZ), ITCZ, AB, and cyclonic vortices high-level electronics (VCANs). These meteorological systems, in turn, undergo alterations when the ENSO phenomenon occurs.

In particular, the position in which AB is located affects the conditions of rain formation in important parts of the Brazilian territory and some of the countries around it. The performance of this meteorological system affects the behavior of ZCIT and VCANs on the continent. Figure 6a shows the climatological position of AB (black dots) from October to December. The positioning of AB during the last three months of 2007 was also verified (red dot). Notably, during 2007, AB remained more to the left of its climatological position. The weekly behavior of ITCZ during the October–November–December quarter of 2007 was also observed (Figure 6b), in addition to the identification of VCANs occurring during the same period (Figure 6c).

*3.4. Weather Simulation (SSPs)*

In the simulation of scenarios SSP2-4.5 and SSP5-8.5, precipitation and ambient temperature showed similar trends. In Figure 7, boxplot graphs for precipitation show the maximum values during March and April and the minimum values during July and August. Regarding ambient temperature, in the boxplots of both scenarios, the maximum values were recorded during September and the minimum values were recorded during June. Outliers were observed during November and December for the SSP2-4.5 scenario and during August for the SSP5-8.5 scenario.

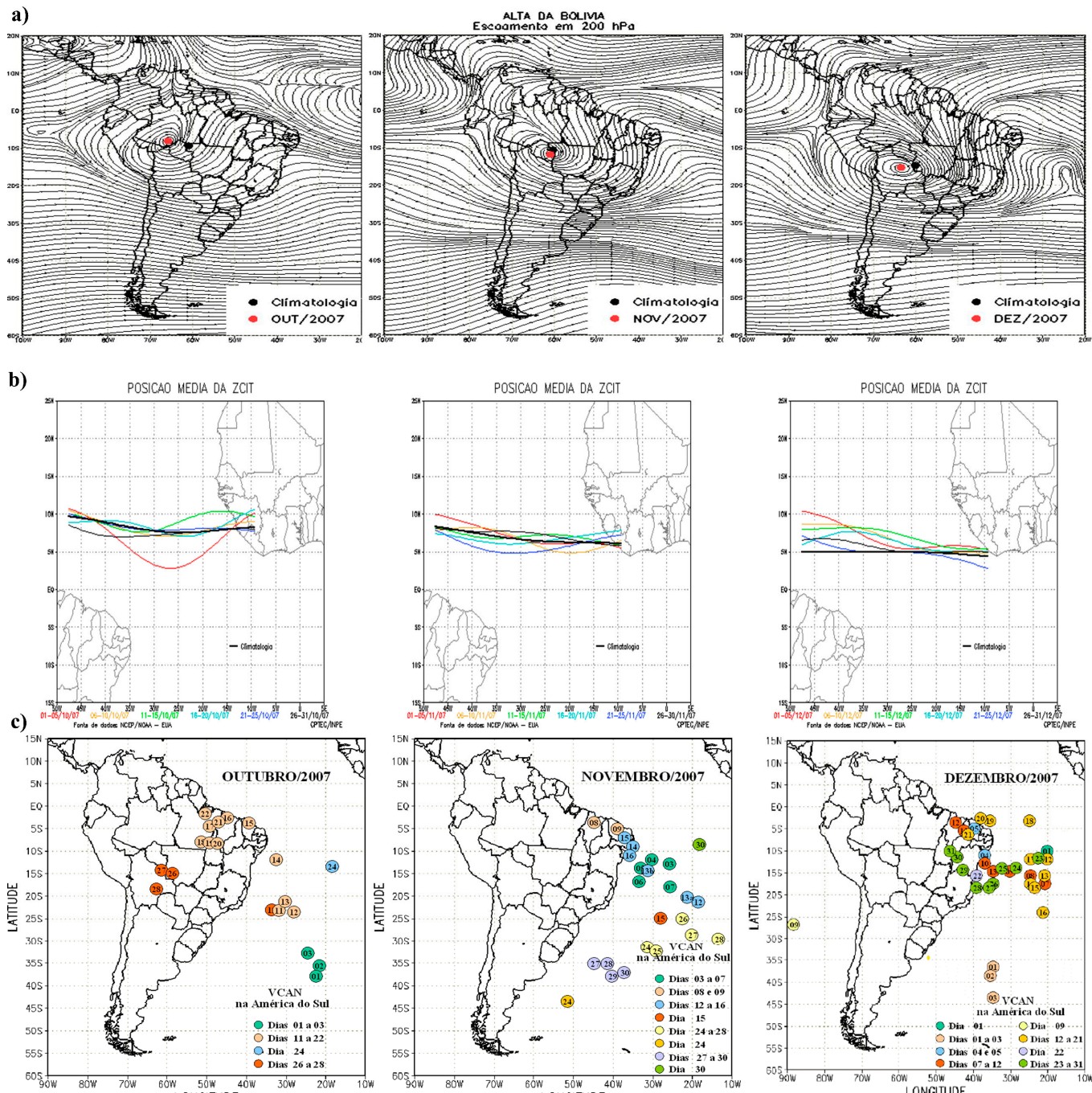

**Figure 6.** (**a**) Position of the AB in the months of October, November and December for 2007. (**b**) Position of the ITCZ in the months of October, November and December for 2007. (**c**) Position of the VCANs in the months of October, November and December for 2007. Source: [36].

All months showed an increase in mean temperature from the scenario SSP2-4.5 to SSP5-8.5. The smallest increase was noted during July, from 24.65 to 24.77 °C (0.45%), whereas the largest increase was noted during December, from 25.69 to 26.21 °C (~2%). Regarding the range of monthly variation, the range expanded from the scenario SSP2-4.5 to SSP5-8.5 in June, but narrowed during December in the other scenarios; the range of variation was larger in the intermediate scenario (SSP2) than in the pessimistic scenario (SSP5). During August and September, when a significant increase was noted in the number of fire foci due to the favorable environmental conditions for the occurrence of fire, the ranges of

ambient temperature were, respectively, 24.15–29.27 °C and 25.63–30.10 °C in the SSP2-4.5 scenario and, respectively, 24.14–30.29 °C and 25.49–31.76 °C in the SSP5-8.5 scenario.

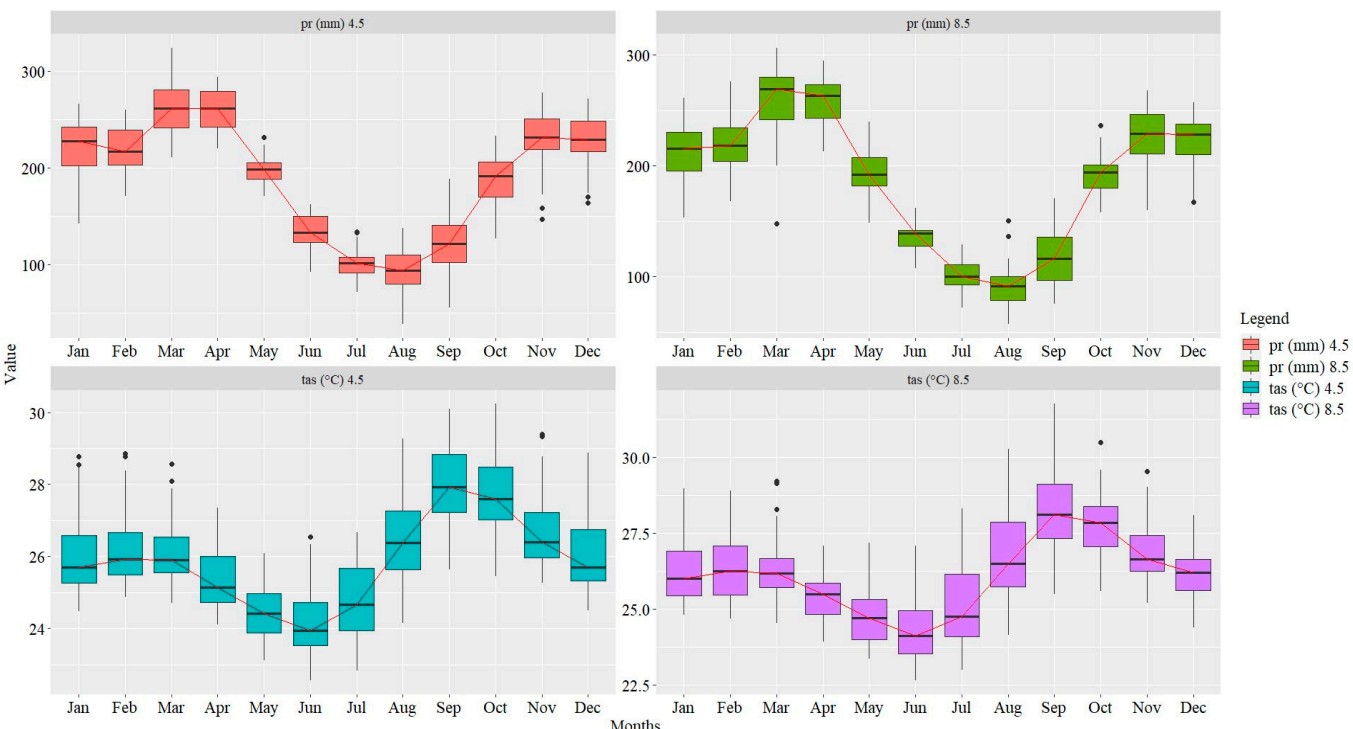

**Figure 7.** Boxplot statistical analysis for precipitation (pr; mm) and air temperature (tas; °C) for future scenarios SSP2-4.5 (**left** column) and SSP5-8.5 (**right** column) of CMIP6 on a monthly scale for the 2015–2050 time series.

Monthly precipitation values both increased and decreased. The largest increase in mean precipitation from the scenario SSP2-4.5 to SSP5-8.5 in the data series was noted in January (~5.36%), and the largest reduction between the same scenarios was noted in June (~4.88%). The lowest values were recorded in August, from 38.0–137.4 mm in SSP2-4.5 to 56.85–150.1 mm in SSP5-8.5. The largest variation was noted in March, from 210.0 to 323.6 mm in SSP2-4.5 and from 147.5 to 306.2 mm in SSP5-8.5 (Figure 7).

### 3.5. Future Simulation and VSM Past–Future Differences

In the future simulation for the DJF quarter, the biome in general presented the largest area in green, with the exception of some regions, such as the northeastern coast of Brazil and the area between the border of Brazil and Venezuela. Intense green coloration was present in the Atlantic Forest in southern Brazil, on the Colombian Pacific Coast, and in the Brazilian states of Mato Grosso and Pará (Figure 8). In the MAM quarter, part of the VSM values decreased in several regions of the continent and increased in the north of Pará and Guianas. In the same quarter, the northeastern coast saw an increase in VSM, as the rainy season occurred during the MAM period; moreover, because of the increase in VSM values between the DJF and MAM quarters, the maximum values were recorded in the JJA quarter (Figure 8).

In the JJA quarter, the orange color was predominant on the map (values of 0.01) in the south of the equatorial forests of Ecuador and Peru, in the south of the Amazon, and in the Atlantic Forest, with the exception of the interior of the Southern Region of Brazil, which remained in a green coloration, representing higher VSM values during this quarter. The highest frequency of minimum VSM values on the continent was evident during the SON quarter, concentrated in the southern and eastern zones of the FRPE in Ecuador, Peru,

Guianas, and the Brazilian Amazon north of the Atlantic Forest. During the DJF and MAM quarters, VSM values in the Peruvian Andes reduced (Figure 8).

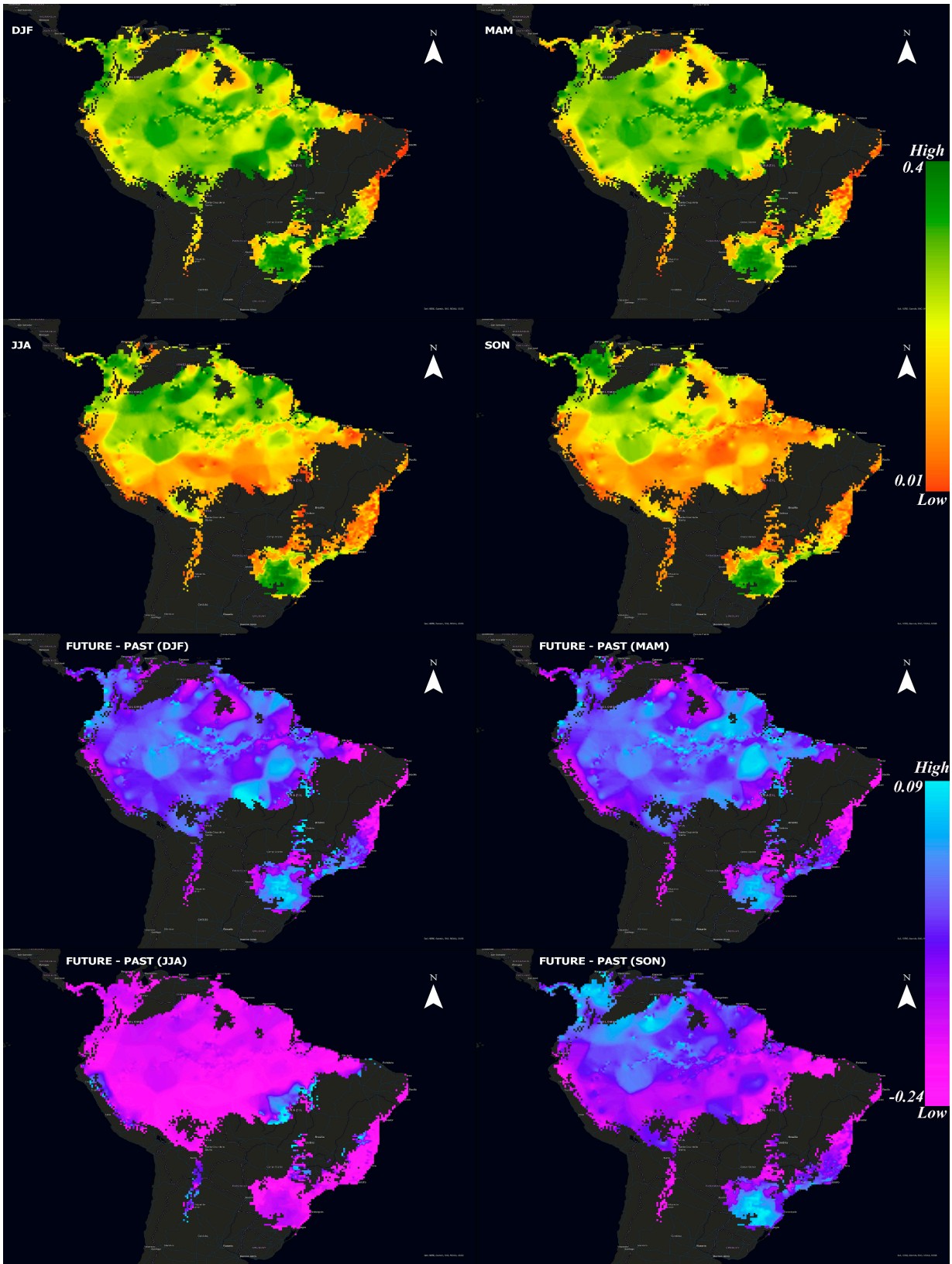

**Figure 8.** Spatial distribution of future VSM (m$^3$ m$^{-3}$) for the 2022–2050 climate quarters and the difference between future and past data.

A future simulation revealed that regions, such as the forest in the interior of the southern region of Brazil, showed high VSM values during all four quarters of the year, as evidenced by the intense green coloration on the map. In the map of differences between the future and past simulations for the JJA quarter, a predominance of the purple color was noted in regions corresponding to the lowest VSM values, representing a marked reduction in water availability in almost the entire biome, with a difference of up to 0.24 m³ m⁻³.

VSM increases were predicted in forests on the North Pacific coast of Colombia and Ecuador. In the Amazon River Basin of Brazil, there was a considerable patch in the state of Mato Grosso in which VSM increased toward the north and decreased to increase again while reaching the state from Pará. Isolated pixels were recorded in the states of Tocantins and Goiás in southern Brazil and particularly in the state of Santa Catarina, Northeast Argentina, with some pixels in the states of São Paulo and Rio de Janeiro.

A reduction in VSM was observed in the Peruvian Andes, Brazilian states on the northeastern coast, around the north of Roraima, in the north of the state of Maranhão on the border with Pará, in the west of São Paulo, and east of Paraguay. During the MAM quarter, the overall trends remained similar to those in the previous quarter; only some regions presented a small reduction in VSM values, such as the south of the Peruvian Andes and north of the Brazilian state of Mato Grosso. During the JJA quarter, almost the entire biome showed a reduction in VSM values, with the exception of small gains in the Peruvian Andes, the Northeast Region of Brazil, the Brazilian state of Mato Grosso, the border of the state of Pará, the north of the Peruvian Andes, and an isolated part of the FRPT located between Bolivia and Argentina. During the SON quarter, VSM increases were noted in the north of the continent in Colombia, south of Venezuela, east of the Amazon, and south of Brazil (Figure 8).

### 3.6. Correlation between the Observed and Simulated VSM Data and ARIMA Analysis

The simulated VSM data were validated based on linear correlation coefficients between the observed and simulated soil water data. A linear trend was noted for the 2001–2021 time series, and the $R^2$ values (available in each year's graph) were in the range of 0.78 (2006 and 2021) to 0.98 (2008 and 2012) (Figure 9). In addition, statistical measures (R, EPE, and D) were calculated to corroborate the linear correlation between the observed and simulated values (Table 2).

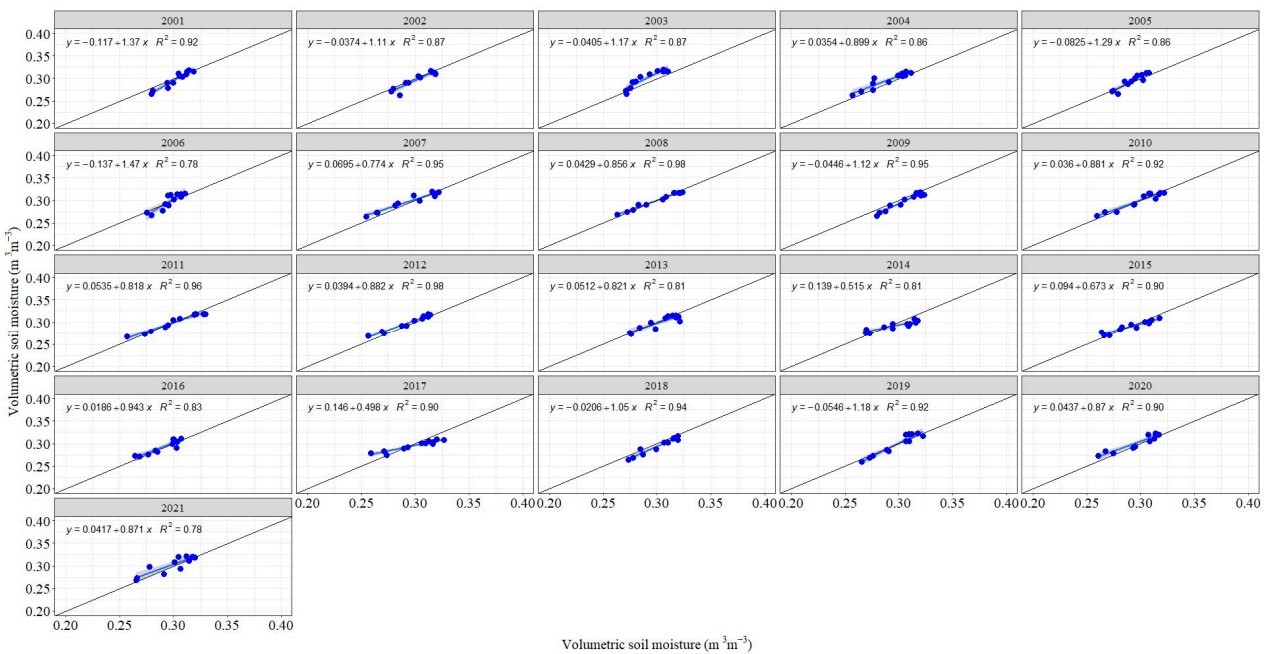

**Figure 9.** VSM regression analysis for observed and simulated data performed by ARIMA modeling (m³ m⁻³) for the years 2001–2021.

**Table 2.** Statistical variables of the correlation of observed and simulated VSM data.

| Year | R | EPE (m$^3$ m$^{-3}$) | D |
|------|-----|-----|-----|
| 2001 | 0.9595 | 0.0053 | 0.9995 |
| 2002 | 0.9315 | 0.0070 | 0.9995 |
| 2003 | 0.9322 | 0.0073 | 0.9990 |
| 2004 | 0.9256 | 0.0071 | 0.9994 |
| 2005 | 0.9297 | 0.0063 | 0.9997 |
| 2006 | 0.8838 | 0.0089 | 0.9994 |
| 2007 | 0.9738 | 0.0047 | 0.9996 |
| 2008 | 0.9900 | 0.0027 | 0.9999 |
| 2009 | 0.9753 | 0.0043 | 0.9995 |
| 2010 | 0.9595 | 0.0056 | 0.9998 |
| 2011 | 0.9780 | 0.0043 | 0.9998 |
| 2012 | 0.9891 | 0.0027 | 0.9998 |
| 2013 | 0.8999 | 0.0072 | 0.9996 |
| 2014 | 0.8993 | 0.0048 | 0.9992 |
| 2015 | 0.9488 | 0.0044 | 0.9996 |
| 2016 | 0.9127 | 0.0066 | 0.9997 |
| 2017 | 0.9474 | 0.0039 | 0.9990 |
| 2018 | 0.9718 | 0.0046 | 0.9996 |
| 2019 | 0.9577 | 0.0070 | 0.9997 |
| 2020 | 0.9502 | 0.0060 | 0.9996 |
| 2021 | 0.8830 | 0.0097 | 0.9994 |

ARIMA modeling follows a seasonal pattern, which involves all values of the data series, including their errors, to correct the behavior of the graph and create a behavior pattern, in this case, for predicting the behavior of future minimum and maximum values from VSM. In Figure 10, the black line corresponds to the observed VSM data, the blue color corresponds to the future simulation for the years 2023–2050 and the light and dark gray intervals represent the potential values that the data could reach. The graph is on a logarithmic scale to facilitate the visual appreciation of the range of future VSM values.

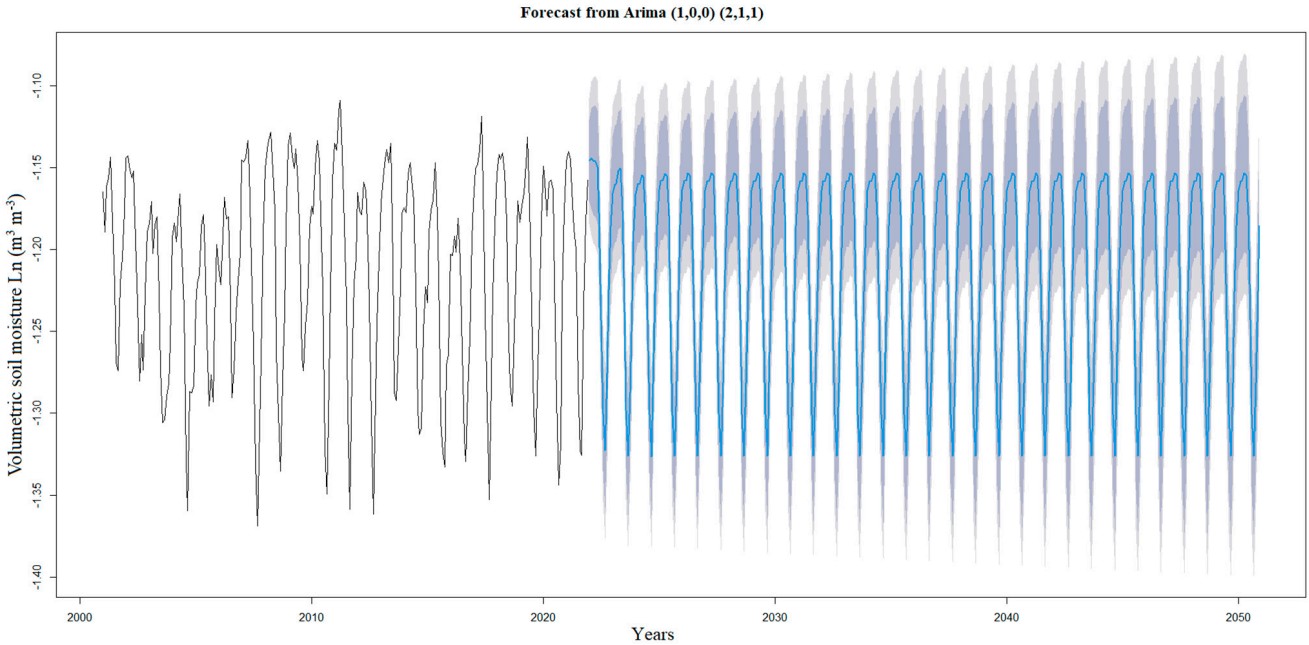

**Figure 10.** Past time series (2001–2021) observed from the VSM (m$^3$ m$^{-3}$) and future time series in blue (2022–2050) through ARIMA modeling. The vertical axis is on a logarithmic scale [12].

### 3.7. Correlation Analysis

In the correlation matrix, the significant values were ambient temperature and "very strong" El Niño (VSE) with a positive Spearman correlation coefficient of r = 0.564. A "very strong" El Niño event increased the ambient temperature in the largest part of the biome located close to the equatorial line, as in the case of the Amazon forest. The hot phase of an ENSO event is a sign of drought and reduced precipitation in the Amazon, and is even more so in the "very strong" phase, when the effects of the phenomenon are more pronounced.

Two negative correlations were observed. First, a negative correlation was noted between VSM and fire foci (r = −0.655); as such, the increase in fire foci, related to the presence of fire, typically of anthropic origin, implies a loss of moisture and decrease in water in the environment, which in turn decreases the VSM in soil pores. Second, ambient temperature was negatively correlated with precipitation (r = −0.757). The role of ambient temperature in rain formation is related to the rise in warm air currents to higher levels, where the pressure is lower, resulting in a decrease in temperature. Thus, air masses reach the dew point temperature of water content in the air, producing rain (Figure 11).

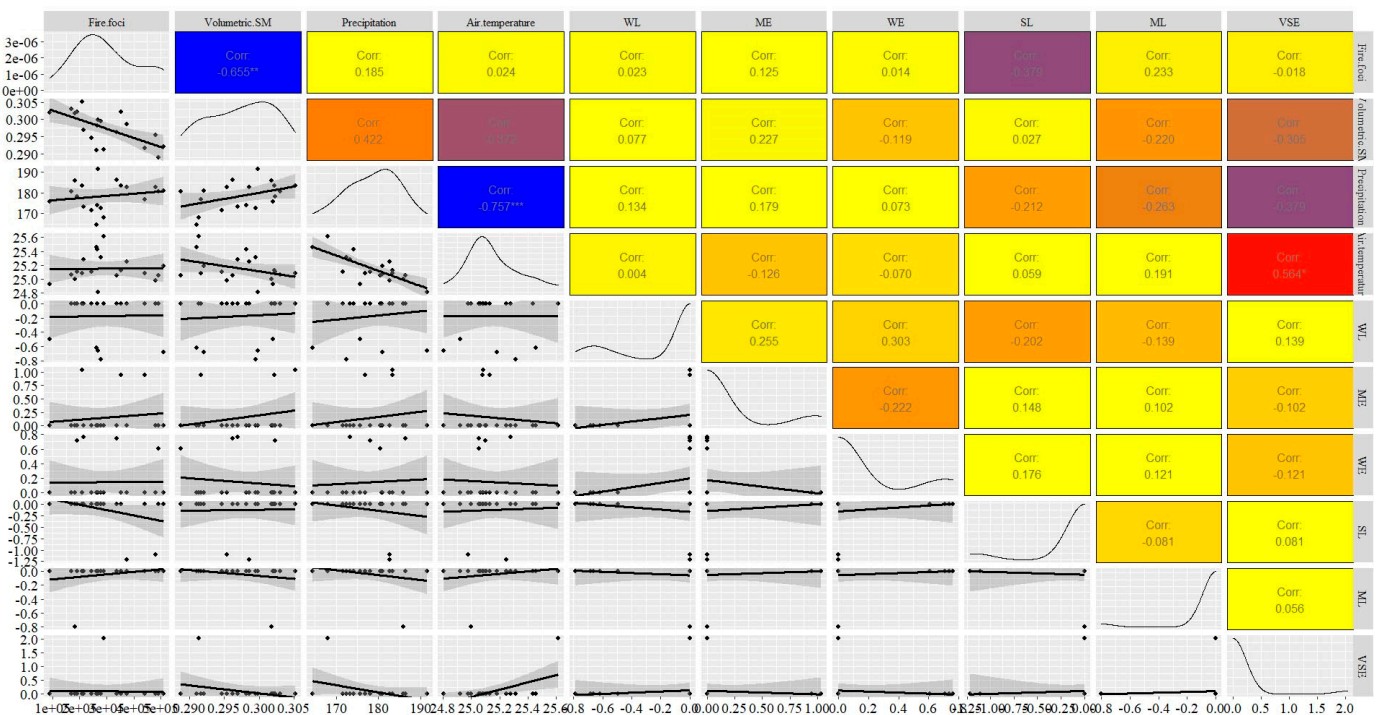

**Figure 11.** Correlation matrix for climate variables. Legend: fire foci, VSM, precipitation, air temperature, "Weak" La Niña (WL), "Moderate" La Niña (ML), "Strong" La Niña (SL), "Weak" El Niño (WE), "Moderate" El Niño (ME), "Strong" El Niño (SE), and "Very Strong" El Niño (VSE) for all years in the period 2001–2019 with the ENOS classification.

### 3.8. Regional Analysis of Acre and Rio de Janeiro

In 2005, the EAC had nearly 35,000 fires, this was the highest number of fires in the time series. More than 90% of the values occurred in the months of September and August. In 2010, an ENSO event was present in the El Niño phase; the number of fire foci was over 15,000, and during the next few years the number of fire foci per year remained below 10,000, only being surpassed in 2016 during the great drought of 2015–2016 and decreasing again in 2017. From this year, the values of fire foci increased each year, surpassing in 2021 the 15,000 fire foci (Figure 12).

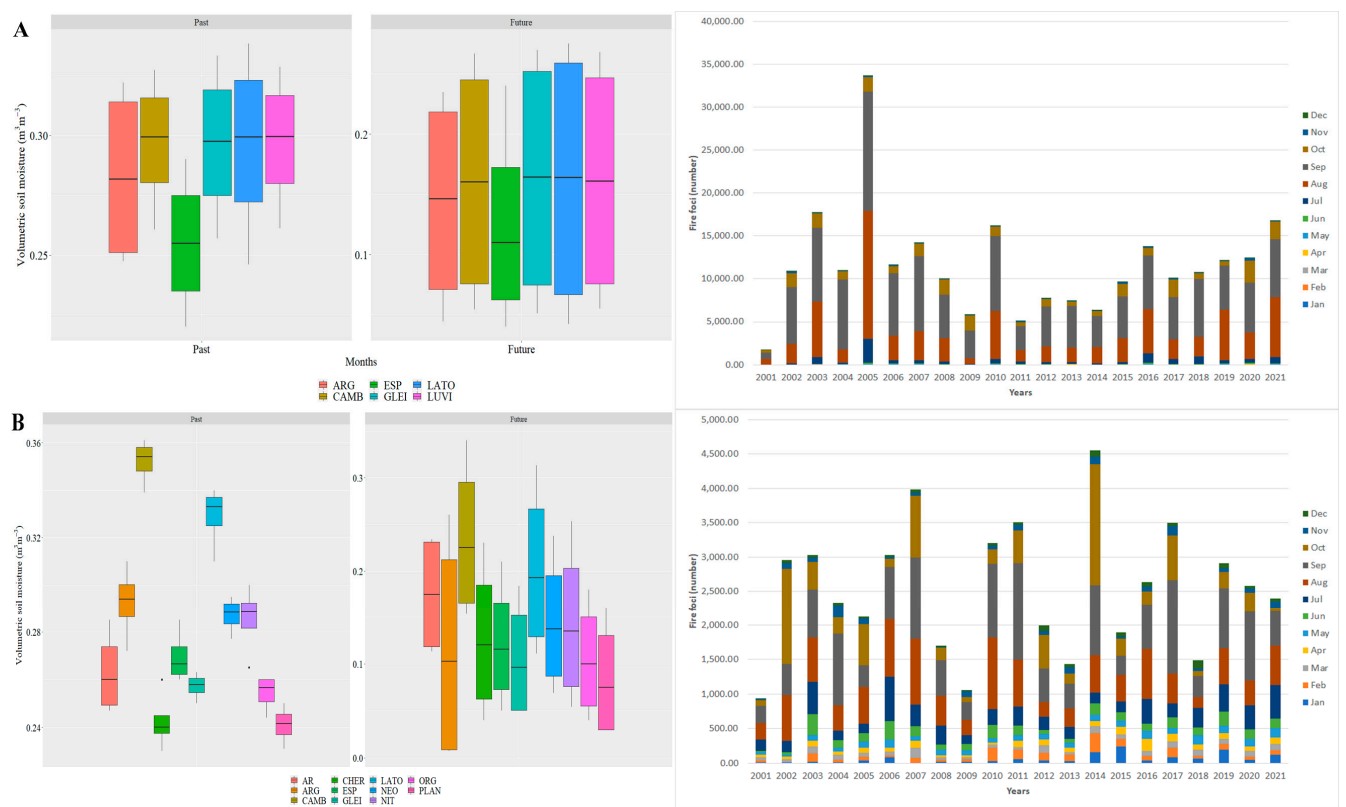

**Figure 12.** Boxplot graphs of VSM variation (m³ m⁻³) for soil types for the states of Acre (**A**) and Rio de Janeiro (**B**) and, on the right, the monthly temporal distribution of fire foci.

The behavior of VSM values in the future simulation for EAC soils showed a significant decrease. The Argissolo and Latossolo units had the highest increase in values in the variation range between soil orders in the future VSM simulation. The Argissolo class went from the range of values between 0.25 (SON) to 0.33 (MAM) m³ m⁻³ in the past period to the range of values between 0.05 (SON) to 0.27 (DJF) m³ m⁻³ in the future simulation, and the Latossolo class obtained a range of values from 0.25 (SON) to 0.34 (MAM) m³ m⁻³ in the past period and a range of values from 0.04 (SON) to 0.28 (DJF) m³ m⁻³ in the future simulation. For the other soil orders present in the EAC, the ranges of VSM values for the past period (2001–2021) and the future simulation (2022–2050) in m³ m⁻³ units were for the Luvissolo class, which were 0.26 (SON) to 0.33 (MAM) in the "Past" period, and 0.05 (SON) to 0.27 (DJF) in the "Future" simulation; for the Cambissolo class, which were 0.26 (SON) to 0.33 (MAM) in the "Past" period, and 0.05 (SON) to 0.27 (DJF) in the "Future" simulation; and for the Gleissolo class, which were 0.26 (SON) to 0.33 (MAM) in the "Past" period, and 0.05 (SON) to 0.27 (DJF) in the "Future" simulation. The Espodossolo order represented the range with the lowest VSM values in the past period with values between 0.22 (JJA) to 0.29 (DJF), obtaining future values in the range of 0.04 (SON) to 0.24 (DJF).

Regarding the ERJ, the years 2001, 2009, 2013 and 2018 showed less than 2000 fire foci per year, while in 2014 the ERJ had the highest number of fire foci in the time series, with 4500 fire foci, followed in descending order by the number of fire foci in 2007, 2011 and 2017 (Figure 12).

The months of August and September represented a significant increase in the number of fire foci. The month of October represented a significant contribution in the years 2002, 2007 and 2014. There has been a reduction in the number of fire foci per year since 2017, and after a significant reduction in 2018, 2021 had the lowest number of fire foci.

For the future modeling of VSM for the ERJ, a reduction in the values of VSM and an increase in the range of variation was observed for all soil units; the same happened with the EAC for the soil classes with the largest area of the state, such as the Cambissolo class

that presented the highest VSM values, with a range between 0.339 and 0.357 m³ m⁻³ in the observed data for the "Past" historical series, while the future simulation maintained the position of the order with the highest values, although the range of future data reduced from 0.15 (JJA) to 0.34 (DJF) m³ m⁻³. For the Argissolo class, there was an increase in the range of values for the future simulation, from 0.29 (SON) to 0.31 (DJF) m³ m⁻³ in the 2001–2021 series to 0.007 (JJA) to 0.26 (DJF) m³ m⁻³ in the series 2022–2050. For the Latossolo class, which also had a significant presence in this state, although it was smaller than those previously mentioned, it presented a smaller difference in the interval of the past period, which was from 0.31 (JJA) to 0.34 (DJF) m³ m⁻³, while in the future simulation there was an increase in the range of values, with a minimum value of 0.11 (JJA) m³ m⁻³ and a maximum of 0.25 (MAM) m³ m⁻³. For the other soil orders, there was less extension in m³ m⁻³ in the in the "Past" period and "Future" simulation, highlighting the quarter of the value obtained. For the Gleissolo class, the values were 0.25 (DJF) to 0.26 (SON) in the "Past" period, and 0.05 (JJA and SON) to 0.18 (DJF) in the "Future" simulation; for the Espodossolo class, the values were 0.26 (MAM) to 0.29 (SON) in the "Past" period and 0.05 (JJA) to 0.15 (MAM) in the "Future" simulation; for the Planossolo class, the values were 0.23 (DJF) to 0.25 (SON) in the "Past" period and 0.03 (JJA) to 0.16 (DJF) in the "Future" simulation; and for the Neossolo class, the values were 0.28 (JJA) to 0.29 (SON) in the "Past" period and 0.07 (JJA) to 0.24 (DJF) in the "Future" simulation.

## 4. Discussion

### 4.1. Climate of the "Tropical Large Leaf Forest" Biome and VSM

The behavior of the maps created based on VSM for Biome 1 showed consistency with the results of the climatic data on air temperature and precipitation. Target areas of a large proportion of studies conducted to date assessing the state of the tropical longleaf forest biomes in AS were located in Brazil [18,19,37–41]. In Brazil, approximately 60% of the Amazon Forest and practically all of the Atlantic Forest constitute most of Biome 1 in AS. Existing research on other forest systems on the continent belonging to this biome is limited, especially in forests located in the northern area of the continent on the Pacific and Caribbean coasts, in the Andes Mountains, and on the border of Bolivia and Argentina.

Regarding the rainfall regime on the continent, in the north of AS, the maximum values of precipitation were recorded in the JJA quarter. Meanwhile, further south in the Northeast, Southeast, and South regions of Brazil, the maximum values of rainfall were recorded in the DJF quarter. As a result of its position close to the equator, the central region of Colombia has a bimodal rainfall regime, with two precipitation peaks during the course of the year, which are more precisely in the months of April and October. In addition to the effects of climate variability on AS, the effects of ENSO phenomena either increase or decrease rainfall according to the geographic location on the continent [42].

In the Amazon rainforest, most of the minerals available in the soil are provided by vegetation through the process of nutrient cycling, which is a process whereby plants return nutrients to the substrate in the form of litter to be reused again by the forest. Large trees transport water confined in underground aquifers through their roots toward the atmosphere through transpiration. Owing to the vast number of trees that the Amazon is home to, a large volume of moisture is turned into rainfall in the region; it is transpired back to the atmosphere by the trees, enabling the recirculation of moisture, which is displaced by the action of wind in the westerly direction, until it collides with the Andes, and by the effect of the mountain range toward the south, it supplies precipitation to the other regions along the way [42–44]. In an ecosystem where more than one-third of the moisture that forms precipitation originates from the evapotranspiration of forest trees, large losses from deforestation lead to significant reductions in the moisture content of air [44].

Although the Amazon is a humid forest, in the recent historical period analyzed (spanning 20 years), there has been evidence of the extension of the dry period in this region; a decrease in rainfall during the flood period suspends plant growth and the recovery of forests after periods of drought or fire associated with it [45]. In particularly,

the FRPE vegetation type is not adapted to withstand prolonged or recurrent periods of drought, and its recovery after such an adverse event can require several years [46]. Although FRPEs are widely studied, research on topics such as evapotranspiration in the region is scarce because of the complexity of the factors involved in its analysis [47]. The proposed methodology of applying VSM assumes an approach of monitoring and measuring the water balance in the tropical forests of AS.

The Atlantic Forest is a water reservoir for three quarters of the Brazilian population, and its water sources supply an important part of the electric power generation system in Brazil, Paraguay, and Argentina (the Itaipú and Yacyretá dams) [48]. What remains of the original Atlantic Forest cover at present is primarily composed of isolated portions of forest, which modifies the climatic dynamics of the ecosystem, such as the survival and free flow of animal species [49]; the size of snake species from the Atlantic Forest was proven to have decreased due to the effects of climate change. Another study [50] analyzed different tree species from the Atlantic Forest under drought conditions, concluding that alterations in the vegetation resulting from the increase in exceptionally dry periods cannot be ruled out, given the lack of centenarian individuals for analysis, a product of historical changes in land use in this region.

In 2005, a period of prolonged drought occurred in the Amazon, which was unrelated to ENSO events. At the time, no evidence of an event of this magnitude was available. In 2010, after a period of intense rain in 2009 that caused disasters in the Amazon region, another drought event of greater intensity than that recorded in 2005 occurred, mainly from March to May, achieving a record of historical lows for rivers in the region [51]. From August 2015 to July 2016, El Niño events occurred in the "very strong" phase, leading to drought in the region of an even greater intensity than that registered in 2005 and 2010. While the drought events in 2005 and 2010 were attributed to an increase in the surface temperature of the Atlantic Ocean, the 2015 event was caused by the warming of waters in the Pacific Ocean [42,52]. The future prediction of a further increase in ambient temperature variation may be related to the increased frequency of extremely high-intensity events linked to the decrease or increase in precipitation.

A dry environment, low humidity, and minimal precipitation contribute to the spread of fire. Trees under water stress are more prone to fires, typically caused by cleaning work for agricultural activities [38]. The constant presence of fire foci in the tropical forest biomes is directly related to the loss of vegetation cover due to the impacts generated by anthropic factors, and the high incidence of these factors can jeopardize the ability of the biome to recover by itself, transforming forests into savannas—the so called effect of "savannation" [53].

Environmental deterioration threatens agricultural yield in AS, which is of global significance to ensure food security and makes an important contribution to the economic sector [39]. A previous study [54] proposed a scenario of increased soybean production in Brazil without expanding the cultivation area by making use of former agricultural lands that are currently not in use or are used for cattle raising in the Brazilian Atlantic Forest, Cerrado, and Pampa biomes. As a result, 162 million tons of soy could be produced in Brazil, without increasing the area of deforestation in the Amazon and reducing the environmental impact by 58% from that that at present.

The COVID-19 pandemic (2019–2021) represented a setback in the preservation of tropical forests in AS due to the increase in the rate of deforestation, increase in illegal mining activities, spread of disease, and attacks on indigenous communities that protect the nature reserves in forests [55–57]. At the beginning of the pandemic in 2019, a 44% increase in greenhouse gas emissions was recorded in the Brazilian territory compared with the value in the previous year (2018), which resulted from forest fires in the Amazon [58]. In terms of fire foci, 2019 saw the highest number of fire foci in the decade for August, which was only lower than the value recorded in 2010 for the same month.

In the Amazon, an increase in precipitation was noted during the dry season in areas where the native cover was removed for approximately a decade; after that time, however,

the results showed a tendency of decreased rainfall [59]. According to a previous study [40], in the southern Brazilian Amazon, when deforestation exceeds a certain limit, a vertiginous reduction in precipitation is observed, granting projections for 2050 of a greater weight to deforestation than to the natural variation in rainfall. The protection of tropical forests, particularly those composed of primary vegetation, ensures the precipitation and supply of ecosystem services [44,60].

### 4.2. VSM Analysis

The VSM provides information on the water status of the soil surface and the atmosphere. Its monitoring makes it possible to monitor various processes such as the water cycle, the coal cycle and the energy generation system. High values of VSM are associated with the healthy metabolism of forests and agricultural crops; the measurement and monitoring of this variable goes hand in hand with measuring the effects of deforestation on the environment, for example, by measuring the evapotranspiration rate [61].

FRPEs provide moisture to FRPTs and other biomes on the continent; the occurrence of precipitation in the Atlantic Forest at certain times of the year supplies an important part of its water requirements for this contribution, and in this Brazilian biome there is a small area of original vegetation cover. In the study carried out by the authors of [52], evidence of a prolonged dry period was found in the quarters of JJA and SON, and the dry season of the years 2000–2019 registered lower values of humidity in more than 16 years compared to the values of the period of 1980–1999 [32].

In the results found in this study, a reduction in VSM values was found, and it was possible to appreciate variations in the VSM maps between the borders of national states, which suggests that there was variation in the levels of preservation of the FRPT (Atlantic Forest). While in the forest of the Southern Region of Brazil, the state of the forest can be classified as being in a good state of conservation, high values of VSM were observed, while in the extension of the forest on the border with Paraguay, the values that were recorded were smaller than the VSM values. For the protection of forests in Southern Brazil, ref [62] refers to the importance of strengthening the legal framework for the protection of these ecosystems, showing that in 2007, only 3.1% of the Araucaria forests were in protection areas. This scenario is not very different today, taking into account that the most recent reserves were created between 2005 and 2006, in the states of Paraná and Santa Catarina. It is also evidenced that the created forest reserve areas were not implemented, or landowners located in these areas increased deforestation as a target of evading plans for environmental preservation.

An analysis carried out in areas of isolated forest and zones that had been deforested for some years showed changes in precipitation that were directly related to a lower rate of evapotranspiration caused by the reduction in native vegetation [43]. With a loss of cover, the rate of surface runoff increases, reducing the permanence of water in the soil, producing excessive dryness of the land in times of low precipitation, causing environmental imbalance, and notably causing the worsened state of health of the remaining vegetation. In a drier atmosphere [43], the loss of the ecosystem's native vegetation, in addition to affecting its functioning, reduces its recovery capacity [52]. From the results of the future simulation of VSM, it is possible to evidence conditions that suggest high vulnerability to drought in the continent in the FRPE in regions that present changes in their native soil cover, such as the Southern Brazilian Amazon, the Ecuadorian and Peruvian Amazon, the Andean forests and central Venezuela, and in the FRPT in the regions of the Atlantic Forest in the Brazilian states Bahia and Paraná and the Southeast Region.

### 4.3. Analysis of 2007 VSM Observed Data

The year 2007 was influenced by ENOS events. This year began in the phase of neutrality that happened and progressed to the decreasing phase of "Weak" El Niño that happened in the last period of 2006. The coloring of the pixels located over the Amazon in Figure 5 for the DJF quarter shows a light green and yellow color, due to the fact that at

the beginning of this year the climate was influenced by an El Niño phenomenon, which changed the usual rainfall regime in the flood season that happens in this period [63].

In the course of 2007, the neutral phase passed into the La Niña phase, which at the end of the year reached the "Strong" La Niña phase [35]. Based on this, the meteorological systems for the last three months of this year were analyzed, at the culmination of spring and early summer in the southern hemisphere.

For the month of October 2007, the AB anticyclonic circulation center (ACC) was located approximately at position 9° S/65° W, in the North and Central-West regions of Brazil, northwest of its climatological position. In November, the AB operated mainly over Bolivia and the states of Rondônia and Mato Grosso, in Brazil, with the CCA located at approximately 10° S/60° W, which is close to what is expected in climatology. For December, AB acted predominantly in Bolivia and on the Central-West Region of Brazil. The CCA was configured at approximately 16° S/61° W, which is close to its climatological position [36] (Figure 6a).

In 2007, the ITCZ position remained close to its climatological position, with weak convective activity for the months of September and October. In December, the ITCZ acted predominantly north of its climatological position, which caused again increase in the precipitation observed in the North Region of Brazil during this period [36] (Figure 6b).

With regard to VCAN events, whose intervention is relevant in generating rainfall on the coast of the Northeast Region of Brazil, during the month of October 2007, four events of this type were recorded on the Brazilian coast, of which two configurations stood out (between the days 11–22 and 26–28), which contributed to the reduction in rainfall recorded this month, which was below the historical average in eastern Brazil. In the month of November, there were eight episodes, of which the first (03 to 07) and the fifth (24 to 28) favored the characterization of the two SACZ events in this month. In December, ten VCAN events took place, operating in the Brazilian Northeast Region and east of the South Atlantic Ocean [36] (Figure 6c).

The configuration of meteorological systems during these months under the influence of La Niña, in addition to causing increases in precipitation in the northwest of the continent and east of the Amazon, surpassed the historical average of rainfall in the Brazilian states of Acre and Amazonas for the month of December [42]. The opposite scenario (of a rainfall below average) occurred in most of the Northeast, Midwest and Southeast regions of Brazil, although the formation of SACZ episodes favored the occurrence of above-average rainfall events as an exception to the general behavior in some locations [36].

### 4.4. Changes in the Behavior of Meteorological Systems

With the warming of the planet, which is accentuated in the extreme north and south, the temperature gradient between the equator and the poles has diminished. This, in addition to rising sea levels, has generated changes in the route and documented speed range of wind currents around the globe, generating repercussions for the normal functioning of rain cycles, increasing the frequency and period of extreme weather events both full and dry. Changes in air currents related to thermal imbalance and changes in meteorological systems are driving changes in the climate [64].

The SACZ, ITCZ, and AB meteorological systems, the VCANs and localtype systems determine the regime and variation in precipitation in the tropical forest areas of the continent [40] during the DJF quarter, a product of the performance of the meteorological systems in this season, and produce a monsoon climate in the Amazon, which generates a significant supply of moisture for the active establishment of the SACZ, and the heating (diabatic) of the Amazon basin establishes the conditions for the establishment of the Bolivian high [42]. The results found in Figure 8 for the coming years predict that areas with the maximum average value of VSM are concentrated mostly in the east and in the central part of the west of the Amazon.

Changes in the position of the Bolivian high and in the movement of the anticyclone can lead to important changes in the occurrence of rainfall in Brazil. Forecasts indicate

that the rise in Bolivia will undergo changes that will affect the rainfall pattern in western Amazonia, increasing the effect of the North Atlantic warming on the hydrological cycle [65].

A study carried out [66] revealed regional changes in the AS and the adjacent south Atlantic Ocean, from a comparison of the averages of the circulation position and thermodynamic variables in the DJF quarter for the periods 1979–1991 and 2005–2014 revealed by the influence of the SACZ strength and position, which are related to the intensification of the South Atlantic subtropical high ASAS. This effect reduces the specific humidity in the South Atlantic, which causes a displacement to the south of the SACZ, reducing the transport of moisture by convection, increasing the local saturation deficit, and contributing to the lower number of days with rain. In the same way, convective development favors extreme precipitation events, which correspond to the observed events. The authors recommend carrying out more in-depth research by expanding the years of the study series, but highlight the significance of these results in the scenario of climate change. In the study by the authors of [45], the previous hypothesis is resumed, relating the effects of the displacement of the SACZ outside Central Brazil, with the acceleration of the hydrological cycle.

In line with this statement in Figure 8, the average values of VSM per pixel showed increases in the VSM of up to 0.09 $m^3$ $m^{-3}$ for the DJF and MAM quarters in the Pacific Coast of Colombia, Eastern Amazonia and the interior of Southern Brazil. This region also recorded increases in the SON period along the north of the continent. In the SON period, the reduction in the VSM in the south of the Amazon and in the Atlantic Forest is highlighted, and the trend of the reduction in the VSM in these regions reached values of up to 0.24 $m^3$ $m^{-3}$ in the JJA period.

### 4.5. ARIMA Modeling

The use of ARIMA modeling in future prediction applications of drought periods [41], through the assessment of the vegetation health index (VHI) for the period 2019–2030 for the Brazilian state of Amazonas, obtained representative results for the region's climate; for the future prediction in the normalized difference moisture index (NMDI) calculation for 12 different land uses in the ERJ, Brazil [17]; for the prediction of areas vulnerable to fires; and for formulating a ranking between 'high' and 'very high' in terms of fire vulnerability for the past and future period in the ERJ [67]. Having identified the months of May and June 2030 as having a high probability of fire occurrence, and the months of August and September as having the highest fire risk values, scientific support was provided for using modeling in the future VSM prediction of "Biome 1".

The results of the ARIMA modeling showed an increase in the variation range of the VSM values in the future period (2023–2050) for Biome 1 (Figure 10). The results found in some scientific works [68,69] predict considerable changes in the climate as known, including the increased occurrence of extreme events. Alternations between extremely wet and extremely dry periods generate health problems in trees, a decrease in the respiration rate, or attacks by pathogens which can lead to the death of the specimens with these changes in the environment [50]. During periods of extreme drought, deaths of trees increase and there is an increase in the occurrence of fires due to the increase in temperatures and the VSM deficit [70]. There is a limitation in quantifying the effects of low humidity resulting from deforestation in the Amazon due to the influence of other relevant factors in the hydrological variation of the Amazon, such as the ENSO phenomenon.

In AS, ENOS considerably alters the climate in most of the continent, even though the operation and occurrence of this phenomenon have been widely studied, and its effects in South America and the rest of the planet [71–74], in the current scenario of global climate change, changes in the environment caused by human action, large-scale changes in land use, have caused repercussions for the intensity and frequency of ENSO phenomena and the information documented in this respect, increasing the impact of extreme events that generate losses of a socioeconomic nature [42,75]. Models close to the real behavior of an ENSO event rule out changes in the known behavior of the phenomenon due to the

influence of the greenhouse effect, but predict an increase in the variability in which it occurs, which suggests an increase in the magnitude of its effects in relation to the increasing rate at which it occurs. El Niño/La Niña events are expected to happen in the "Very Strong" phase [42]. Therefore, the future prediction of this phenomenon and the relationship of its effects on meteorological systems become relevant in the search for actions to mitigate the degradation of tropical forests due to anthropic causes and the effects of climate change [46].

*4.6. Local Analysis Acre and Rio de Janeiro*

For both EAC and ERJ, the VSM variation range increased in the future simulation, which was related to changes in evapotranspiration, runoff, precipitation and other factors linked to the VSM variation, increasing the impact of extreme rainfall and the impact generated by floods, just as in the dry period, drought and the water supply cause more significant damage, adding to the area of impact of the effects of climate change on a global scale.

The year 2004 had the highest number of fire foci in the EAC, and this was also the year with the highest rate of deforestation in the legal Amazon [76]. In 2005, eastern Amazonia experienced a severe drought, the conditions of which were conducive to the generation and propagation of fires related to agriculture and cattle raising. In the results of this study, a significant increase was found in the number of heat foci for this year in the time series.

The current context of deforestation and change in land use in the EAC began in the 1970s when the government at the time promoted a policy of integrating the Amazon with the rest of Brazil through the construction of roads. At that time, the economy of rubber and chestnut (important economic sectors in the EAC) had a setback, in response to which the state government chose to direct the economy of the cattle raising sector in the Amazon through incentives. This policy attracted settlers from the center-south of Brazil, which, in addition to generating new social problems, increased the area of deforestation [77].

Based on the results found in this study, there is evidence of a tendency towards an increase in the number of fire foci in the EAC in recent years (Figure 12). In this region this is related to land use change activities, such as loss of original vegetation cover. In the EAC, the municipalities with the greatest length of roads have a higher rate of deforestation, due to the ease provided by roads of accessing and extracting resources, and cleaning and acquiring land. This situation occurs throughout the Amazon, on roads built under a legal or illegal framework, and represents a challenge for the exercise of territorial control for the protection of the environment [78–80]. In 2017, the deforested area in the EAC was 257 km$^2$, and in 2018 there was an increase of 82.9% in this measurement referring to the previous year, registering an area of 470 km$^2$ [81].

The soils in the EAC stand out among the soils of the Brazilian Amazon due to their pedogenesis, marked by their geographic proximity to the Andes Mountain range across the border with Bolivia and Peru, which is attributed to the Solimões Formation and the contribution of sediments in its composition, present in the order Cambissolos, characterized by the fertility of their soils, despite the limitations in structure or the erosion typical of the relief in which they can be found. In some types of soils in this region, the minimal presence of clays is associated with weathering from the Andes [82]. The reduction in the VSM content of all soil classes in the future simulation could aid the existing vegetation in a situation of stress, in soils of the order Argissolo, which present an increase in the content of clays in the lower layers, or soils of the order Luvissolo which have natural fertility, despite presenting agricultural limitations due to their high susceptibility to erosion, the presence of expansive clays and poor drainage that restricts the establishment of some crops [83]. Likewise, the increase in the range of variation in VSM values may be related to the occurrence of extreme drought and flood events.

The ERJ is a densely populated region, and is even more so when compared to the EAC. Historically, this territory has experienced many changes in land use, a product of human flows, and political and social dynamics that have taken place for centuries in this place, which currently exemplify the situation of large populated areas that have

developed close to the areas of mountain forests, which puts them at high risk of anthropic interventions. The supply of water resources for almost the entire population of the ERJ and millions of people in the surrounding states comes from the tropical mountain forests Serra do Mar and Serra da Mantiqueira [84], through which the preservation of the Atlantic Forest is relevant topic on the political agenda, including the approach towards taking measures to face savannization and the effects of climate change.

In the last three years, there has been a continuous decrease in the number of fire foci in this state, which due to its geographical location on the coast, is susceptible to high-speed winds, high air temperatures with a forecasted increase and low humidity, which generate conditions for the appearance of fires and that favor their propagation.

The past simulation of VSM in the existing soil orders in the ERJ made it possible to observe varied behavior among the existing soil orders in this state. The highest representativeness of soils corresponds to the orders Argissolo, Espodossolo and Gleissolo (Figure 1). Argissolos are soils with clearly differentiated horizons, a clay content of 50% more in the subsurface horizon than in the horizon above it, variable depth, imperfect drainage and strong to moderate acidity. The Espodossolos order has the greatest sandy texture among the orders, but it has the lowest water retention, explaining the reason for the low VSM values in the EAC and ERJ results; this order also has low-fertility soils, low base saturation and low cationic capacity, a characteristic that is also evident in the order Gleissolos, although the soils of this order are considered of good fertility in terms of performing corrections for the exercise of agriculture [18].

Considering the future simulation of ground orders for the ERJ, it was possible to perceive a significant reduction in VSM values for all ground orders and a notable difference in the ranges of VSM values between past and future simulations. After finding in the past period different patterns between the quarters of the occurrence of the minimum and maximum values in the different soil orders, during the future simulation, all soil orders showed a minimum value of VSM in the autumn SON quarter and a maximum value during the autumn quarter.

This increase in the range of future VSM values can be related to changes in known weather patterns and an increase in extreme events, based on the characteristics of soil composition and changes in the existing land use in this region [15].

In Atlantic Forest restoration initiatives (the Brazilian biome that has undergone the greatest alteration in the original use of the soil), it is recommended carry out soil identification in the restoration plot [85]. In addition to scientific and financial support, strengthening the legislative and criminal framework that involves the preservation of tropical forest areas in South America is recommended [57].

## 5. Conclusions

The present study aimed to integrate an ARIMA analysis of VSM with climate data as an environmental diagnosis methodology for the formulation of strategies contributing to the knowledge on the large leaf tropical forest biomes in AS. In addition, a future forecast for 2050 was presented, seeking to contribute to decision making in the face of climate change driven by anthropic action, primarily land use changes, which have contributed to the increased frequency and severity of extreme weather events, altered trends of meteorological systems, and increased intensity and frequency of ENSO phenomena.

The months of August and September registered the highest values of fire foci for the entire biome. The future increases in ambient temperature and decreases in precipitation may be related to the extension of dry periods. In future modeling with SSPs, a ~5.36% increase in precipitation during January and a ~4.88% reduction in precipitation during June are expected throughout the biome between the intermediate (SSP2-4.5) and pessimistic (SSP5-8.5) scenarios.

Furthermore, according to the ARIMA analysis, the maximum change of 0.24 $m^3$ $m^{-3}$ for the coming years until 2050 and an increase in the variation range of VSM values are expected. The DJA quarter showed the highest VSM values in proportion to the

biome area, whereas the JJA quarter showed the lowest values. Among the regions of the biome showing the lowest VSM values, the southern Amazon (Ecuador, Peru, and the Brazilian states of Acre, Mato Grosso, Pará, and Maranhão), the Brazilian Atlantic Forest, the Southeast Region, and the Brazilian State of Bahia stand out, which harbor the largest portions of this biome in the northeast region. Local analysis showed a progressive increase in the number of fire foci for EAC in recent years, whereas a slight gradual reduction in ERJ was noted over the last 3 years. VSM values decreased in all soil classes.

**Author Contributions:** Conceptualization: S.M.M.A. and R.C.D. Formal analysis: S.M.M.A., R.C.D., D.d.S.L., Y.A.G., M.G.P., R.d.Á.R., F.B.J., H.S.W., E.Z., R.O.d.S. and R.S.d.S. Resources: R.C.D. Data curation: S.M.M.A., R.C.D., D.d.S.L., Y.A.G., M.G.P., R.d.Á.R., F.B.J., H.S.W., E.Z., R.O.d.S. and R.S.d.S. Writing—original draft preparation: S.M.M.A., R.C.D., D.d.S.L., Y.A.G., M.G.P., R.d.Á.R., F.B.J., H.S.W., E.Z., R.O.d.S. and R.S.d.S. Writing—review and editing: S.M.M.A., R.C.D., D.d.S.L., Y.A.G., M.G.P., R.d.Á.R., F.B.J., H.S.W., E.Z., R.O.d.S. and R.S.d.S. Visualization: S.M.M.A., R.C.D., D.d.S.L., Y.A.G., M.G.P., R.d.Á.R., F.B.J., H.S.W., E.Z., R.O.d.S. and R.S.d.S. Supervision: R.C.D. Project administration: S.M.M.A. and R.C.D. All authors have read and agreed to the published version of the manuscript.

**Funding:** This research was funded by the Research Support Foundation of the State of Rio de Janeiro—FAPERJ grant (201.188/2022), and the National Council for Scientific and Technological—CNPq grant (308922/2021-2).

**Institutional Review Board Statement:** Not applicable.

**Informed Consent Statement:** Not applicable.

**Data Availability Statement:** Not applicable.

**Acknowledgments:** We also thank the Federal Rural University of Rio de Janeiro, Federal University of Viçosa and Higher Education Personnel Improvement Coordination—CAPES.

**Conflicts of Interest:** The authors declare no conflict of interest.

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
