# Peer review of "Past and Future Responses of Soil Water to Climate Change in Tropical and Subtropical Rainforest Systems in South America"

_atmosphere, doi:10.3390/atmos14040755_

Round 1

Reviewer 1 Report

the paper is well prepared, but the results seem obscure. Many statistical tests are shown, but its difficult to find an emerging trend, or clear link to ENSO etc. Some proof-reading is needed, eg. line 762 'rise in Bolivia..' Fig 10 shows past and future soil moisture, but future does not show variability or trend, mainly increase of seasonality: drier winters, wetter summers. The reviewer did a brief analysis using area A, and found a drying trend in ISIMIP2 forecasts using HADGEM rcp6, but little trend in FLDAS in area A. A map of trends was made, and lag-correlations with ENSO were done, perhaps the authors could add these type of analysis, to obtain a clearer picture. The authors seem to concentrate on individual years which tends to disperse the focus on trends and regional variability in uptake of global warming signals. One question is whether the future drying trends are linked to potential evaporation or rainfall? The authors could do a trend analysis on ERA5 geopotential height (1950-2022), to understand whether the Bolivian high is expanding? or whether trends near the Pacific differ from those near the Atlantic, and why? presumably it is more than just ENSO, but related to westward moisture fluxes. These are only a few ideas to add to the paper, so to help the authors focus better on climate change outcomes...

Author Response

Reviewer #1

the paper is well prepared, but the results seem obscure. Many statistical tests are shown, but its difficult to find an emerging trend, or clear link to ENSO etc. Some proof-reading is needed, eg. line 762 'rise in Bolivia..' Fig 10 shows past and future soil moisture, but future does not show variability or trend, mainly increase of seasonality: drier winters, wetter summers. The reviewer did a brief analysis using area A, and found a drying trend in ISIMIP2 forecasts using HADGEM rcp6, but little trend in FLDAS in area A. A map of trends was made, and lag-correlations with ENSO were done, perhaps the authors could add these type of analysis, to obtain a clearer picture. The authors seem to concentrate on individual years which tends to disperse the focus on trends and regional variability in uptake of global warming signals. One question is whether the future drying trends are linked to potential evaporation or rainfall? The authors could do a trend analysis on ERA5 geopotential height (1950-2022), to understand whether the Bolivian high is expanding? or whether trends near the Pacific differ from those near the Atlantic, and why? presumably it is more than just ENSO, but related to westward moisture fluxes. These are only a few ideas to add to the paper, so to help the authors focus better on climate change outcomes...

We really appreciate the suggestions and understand the relevance of your appreciations, to explore them, and apply in a subsequent paper, going deeper into the behavior of the Bolivian high related with the continent scale climate. We are preparing a second manuscript, I believe that this information and recommendations are very welcome, but due to the time, we would not be able to analyze and carry out any kind of survey on what was mentioned. Despite not showing such a large fluctuation in some climatic trimesters, in the JJA the soil moisture changes abruptly with significant reductions in both sectors, which could potentiate the fires in the Brazilian Atlantic Forest and Amazonia. Once again we greatly appreciate your inquiries and as mentioned above, we will be doing these analyzes more thoroughly and checking other weather systems that may have a greater correlation with fluctuations in temperature and precipitation over the years.

Reviewer 2 Report

I have reviewed the document "Past and future responses of soil water to climate change in tropical and subtropical rainforest systems in South America", prior to further processing it is necessary to review the following:

1. It is important to add one or two introductory sentences in the summary that I narrated about climate change and its problems and then describe the objective.

2. Do not repeat words from the title in keywords, the importance of selecting different words implies greater chances of being found and displayed in search engines. And therefore more likely to be cited.

3. The introduction is very well designed and presents an orderly review of the research variables.

4. The methodology is very well described and presents robustness for the analyses.

5. The presentation of the results is of high quality and is in accordance with the stated objectives.

6. The discussion is very extensive and should be simplified considering the objectives of your study, try not to repeat what is stated in the results.

7. The conclusions should present the possible complications that occurred in the research and future studies based on your results.

Author Response

Reviewer #2

I have reviewed the document "Past and future responses of soil water to climate change in tropical and subtropical rainforest systems in South America", prior to further processing it is necessary to review the following:

  1. It is important to add one or two introductory sentences in the summary that I narrated about climate change and its problems and then describe the objective.

We appreciate the suggestion and understand the importance of this point. A sentence was added to the summary to relate the climate change problem with the objective of the research.

  1. Do not repeat words from the title in keywords, the importance of selecting different words implies greater chances of being found and displayed in search engines. And therefore more likely to be cited.

We appreciate the suggestion and understand the importance of this point, instead of “climate change”, we put the keyword “ARIMA analysis”

  1. The introduction is very well designed and presents an orderly review of the research variables.

Thank you so much for your appreciation.

  1. The methodology is very well described and presents robustness for the analyses.

Thank you so much for your appreciation.

  1. The presentation of the results is of high quality and is in accordance with the stated objectives.

Thank you so much for your appreciation.

  1. The discussion is very extensive and should be simplified considering the objectives of your study, try not to repeat what is stated in the results.

We appreciate the suggestion and understand the importance of this point. The text in the discussion was reduced, we seek to simplify the text, removing parts that can be understood in the results.

  1. The conclusions should present the possible complications that occurred in the research and future studies based on your results.

We appreciate the suggestion and understand the importance of this point,the conclusion was redirected, we hope that this work will contribute to the investigation of climate impact in the tropical forests of South America and in the meteorological systems that act in the continent.

Reviewer 3 Report

The study on "Past and future responses of soil water to climate change in tropical and subtropical rainforest systems in South America" is a interesting work which builds relationship between forest biome, volumetric soil moisture (VSM) and weather parameters under future climate scenarios. Following suggestions are given for better understanding by the readers: 

1. How the El Niño and La Niña events/ years were picked up? Whether such events/ years are also reported elsewhere?

2. How the classes as given in the Table 1. El Niño and La Niña years and intensities were categorised e.g., El Niño [Weak - 11, Moderate - 2, Very Strong - 1] and La Niña [Weak - 6,  Moderate - 3, Strong - 2] ?

3. How the fire foci per month were computed? What do the the numerical values indicate e.g., September, 147,664 ? [Pg. 6, line 221: Almost all years in the data series recorded the highest monthly values during August and September. In 2004 (September, 147,664), 2005 (August, 148,834), 2007 (September, 173,500), and 2010 (September 121,958), the highest values of fire foci per month…]

4. Pg. 18, line 537: The maps created based on VSM for Biome 1 were consistent with the climatic data of ambient temperature and precipitation and the occurrence of ENSO. [Are these ambient temperature or land surface temperature? How such conclusion has been arrived at?]

5. Kindly produce a map showing rainfall, temperature, VSM for corresponding period to demonstrate their identical pattern.

Author Response

Reviewer #3

The study on "Past and future responses of soil water to climate change in tropical and subtropical rainforest systems in South America" is a interesting work which builds relationship between forest biome, volumetric soil moisture (VSM) and weather parameters under future climate scenarios. Following suggestions are given for better understanding by the readers:

  1. How the El Niño and La Niña events/ years were picked up? Whether such events/ years are also reported elsewhere?

We really appreciate the suggestion and understand the importance of this point. These years were selected because of the intensity evidenced by El Niño and La Niña events. The information of El Niño and La Niña events was taken from ggweather.com.

  1. How the classes as given in the Table 1. El Niño and La Niña years and intensities were categorised e.g., El Niño [Weak - 11, Moderate - 2, Very Strong - 1] and La Niña [Weak - 6, Moderate - 3, Strong - 2] ?

We really appreciate the suggestion and understand the importance of this point. The categorization chosen was that of the on the Oceanic Niño Index (ONI), which classifies La Niña and El Niño events according to their intensity. The number on the right represents the number of events of this type that occurred during the analysis period.

  1. How the fire foci per month were computed? What do the the numerical values indicate e.g., September, 147,664 ? [Pg. 6, line 221: Almost all years in the data series recorded the highest monthly values during August and September. In 2004 (September, 147,664), 2005 (August, 148,834), 2007 (September, 173,500), and 2010 (September 121,958), the highest values of fire foci per month…]

The number of fire foci by year were spatialized in GIS software, and subsequently totaled by month using a spreadsheet. August and September were the months with the highest number of fire foci, and the years mentioned were those with the highest values of the entire time series. 

  1. Pg. 18, line 537: The maps created based on VSM for Biome 1 were consistent with the climatic data of ambient temperature and precipitation and the occurrence of ENSO. [Are these ambient temperature or land surface temperature? How such conclusion has been arrived at?]

We really appreciate the suggestion and understand the importance of this point. The temperature mentioned is air temperature above 2 meters. We came to the appointed  conclusion linking the ENSO occurrence, which produces an increase or decrease (depending the phase) in surface water temperature of the Eastern Pacific Ocean, with the increase in air temperature and precipitation reduction in a considerable area of the tropical rainforests located near the Equator.

  1. Kindly produce a map showing rainfall, temperature, VSM for corresponding period to demonstrate their identical pattern.

We really appreciate the suggestions and understand the relevance of your appreciations,  to explore them, and apply in a subsequent paper. Our manuscript presents numerous figures and consequently one more Figure, would greatly increase our results, consequently we are accepting your recommendation and we will be submitting a second manuscript presenting new results and perhaps inserting this new information. Thank you again.

Reviewer 4 Report

The authors used detailed data to illustrate how the soil water response to climate change in tropical and subtropical rainforest systems in South America. The text of the drawings in the manuscript is easy to read, and the results are explained in detail. The manuscript is interesting, but it's just too long and with so many paragraph. Only a few small suggestions are shown below.

1. There are too many preamble paragraphs, which can be appropriately combined through logical language. 

2. The results parts was clearly showed by the beautifully images and it was also very intuitive to see what the authors wanted to express. Therefore, the text part can be appropriately selected and explained to reduce the text.

Author Response

Reviewer #4

The authors used detailed data to illustrate how the soil water response to climate change in tropical and subtropical rainforest systems in South America. The text of the drawings in the manuscript is easy to read, and the results are explained in detail. The manuscript is interesting, but it's just too long and with so many paragraph. Only a few small suggestions are shown below.

  1. There are too many preamble paragraphs, which can be appropriately combined through logical language.

We appreciate the suggestion and understand the importance of this point. The text was reduced mainly in the discussion, other parts were rewritten to summarize ideas and make the document easier to read.

  1. The results parts was clearly showed by the beautifully images and it was also very intuitive to see what the authors wanted to express. Therefore, the text part can be appropriately selected and explained to reduce the text.

Thank you so much for your appreciation. The text was reduced mainly in the discussion, seeking to express specific ideas succinctly.
